



# On the occurrence of enhanced vertical wind shear in the tropopause region: A ten year ERA5 northern hemispheric study

Thorsten Kaluza[1], Daniel Kunkel[1], and Peter Hoor[1]

[1]Institute for Atmospheric Physics, Johannes-Gutenberg University Mainz, Mainz, Germany

**Correspondence:** Thorsten Kaluza (kaluzat@uni-mainz.de)

**Abstract.** A climatology of the occurrence of enhanced wind shear in the UTLS is presented, which gives rise to define a tropopause shear layer (TSL). Enhanced wind shear in the tropopause region is of interest because it can generate turbulence which can lead to cross-tropopause mixing. The analysis is based on ten years of daily northern hemispheric ECMWF ERA5 reanalysis data. The vertical extent of the region analysed is limited to the altitudes from 1.5 km above the surface up to 25 km, to exclude the planetary boundary layer as well as enhanced wind shear in higher atmospheric layers like the mesosphere/lower thermosphere. A threshold value of $S_t^2 = 4 \cdot 10^{-4}\ \mathrm{s}^{-2}$ is applied, which marks the top end of the spectrum of atmospheric wind shear to focus on situations which cannot be sustained by the mean static stability in the troposphere according to linear wave theory. This subset of the vertical wind shear spectrum is analysed for its vertical, geographical, and seasonal occurrence frequency distribution. A set of metrics is defined to narrow down the relation to planetary circulation features, as well as indicators for momentum gradient sharpening mechanisms.

The vertical distribution reveals that large shear values occur almost exclusively at tropopause altitudes, within a vertically confined layer of about 1–2 km extent directly above the local lapse rate tropopause (LRT). The TSL emerges as a distinct feature in the tropopause-based 10 year temporal and zonal mean climatology, spanning from the tropics to latitudes around 70° N, with average occurrence frequencies of the order of $1\,\% - 10\,\%$. The horizontal distribution of the tropopause based enhanced vertical wind shear exhibits distinctly separated regions of occurrence, which are generally associated with jet streams and their seasonality. At midlatitudes, enhanced wind shear values occur most frequently in regions with an elevated tropopause and at latitudes around 50° N, associated with jet streaks within northward reaching ridges of baroclinic waves. At lower latitudes in the region of the subtropical jet stream, which is mainly apparent over the East Asian continent, the occurrence frequency of enhanced tropopause-based wind shear reaches maximum values of about 30 % during winter and is tightly linked to the jet stream seasonality. The interannual variability of the occurrence frequency for enhanced wind shear might furthermore be linked to the variability of the zonal location and strength of the jet. The east-equatorial region features a bi-annual seasonality in the occurrence frequencies of tropopause based enhanced vertical wind shear. During the summer months, large areas of the tropopause region over the Indian ocean are up to 70 % of the time exposed to large values of wind shear, which can be attributed to the emergence of the tropical easterly jet. During winter, this occurrence frequency maximum shifts eastward over the maritime continent, where it is exceptionally pronounced during the 2011 la Niña year, as well as quite weak during the El Niño phases of 2010 and 2015/2016. This agrees with the atmospheric response of the Pacific Walker circulation cell in the ENSO ocean-atmosphere coupling.



# 1 Introduction

The distribution of vertical wind shear in the atmosphere is an substantial feature of the thermodynamic structure because it controls the dynamic stability of the flow. Vice versa, dynamic stability limits the amount of wind shear that is sustainable by the flow. The dynamic stability of the tropopause region is ultimately of interest because turbulent mixing in consequence of dynamic instability modifies the trace gas gradients at the tropopause, which in return can have a significant effect on the radiative budget not only locally but also at the Earth's surface (Forster and Shine, 1997; Riese et al., 2012). Turbulent mixing

at the tropopause is furthermore a pathway for stratosphere-troposphere exchange (STE, Holton, 1995; Stohl et al., 2003), which is important because of to the impact on the chemical budget in the troposphere and the stratosphere.

According to linear wave theory, the dynamic stability of a medium in a stratified shear flow can be evaluated on the basis of the non-dimensional Richardson number $Ri = N^2/S^2$. It is defined as the ratio between static stability $N^2 = g/\Theta \cdot (\partial\Theta/\partial z)$ and squared vertical shear of the horizontal wind $S^2 = (\partial u/\partial z)^2 + (\partial v/\partial z)^2$, with the gravitational accelaration g and the

zonal and meridional wind components $u$ and $v$. The static stability $N^2$ is determined through the vertical gradient of the potential temperature $\Theta = T(p_0/p)^{R_d/c_p}$, with the temperature $T$, the atmospheric air pressure $p$ and a reference pressure $p_0$, and the ratio between the specific gas constant for dry air and the specific heat capacity $R_d/c_p = 0.286$. The Richardson number describes the ratio of the suppression of turbulent kinetic energy due to buoyancy and the production of turbulent kinetic energy due to shear forces. If the flow exhibits Richardson numbers below the critical threshold value $Ri_c = 1/4$, it

can become dynamically unstable (Miles, 1961). A common descriptive example for this issue is the occurrence of Kelvin Helmholtz instabilities (KHI), where the shear-induced horizontal shift of a vertically displaced air parcel results in local convective instability in the elsewhere stably stratified flow, and eventually in flow overturning, turbulent breakdown, and gradient erosion.

The lapse rate tropopause (LRT) is defined as the first vertical level above the surface at which the temperature lapse rate

falls below 2 K km$^{-1}$ and its mean between this level and any level up to 2 km above remains below that threshold (WMO, 1957). This increase in stratification implicates an increase in static stability, from mean upper tropospheric values on the order of $\overline{N^2}_{trop} \approx 1 \cdot 10^{-4}$ s$^{-2}$ to average lower stratospheric values of $\overline{N^2}_{strat} \approx 4 \cdot 10^{-4}$ s$^{-2}$. The vertical profile of $N^2$ furthermore exhibits a vertically confined maximum within the first few kilometres above the LRT which is caused by a localised temperature inversion. This feature is quasi-ubiquitous on planetary and climatological scales, and is referred to as

the tropopause inversion layer (TIL, Birner, 2006). The increase in static stability at the LRT in general, as well as the $N^2$ maximum within the TIL in particular define a background state which can sustain larger vertical wind shear $S^2$ compared to the troposphere. This fact is reflected in the vertical distribution of $S^2$.

Early approaches towards assessing statistical vertical wind shear occurrence frequencies were performed by Dvoskin and Sissenwine (1958), in the context of preparing guidelines for missile design, using a set of radiosonde measurements mostly





taken during winter and spring at the East coast of the USA (41° N–45° N). The data showed distinct occurrence frequency maxima for enhanced values of $S^2$ within sampling windows of 3000 ft at altitudes between 30000 and 40000 ft.

The occurrence of enhanced $S^2$ in the tropopause region was described in several research studies which analyse vertical profiles of atmospheric flow properties in a vertical coordinate system relative to the LRT altitude (Birner, 2006). In the pioneering work on the existence of the TIL, Birner et al. (2002) describe a sharp peak of $S^2$ at the LRT in tropopause-based

averaged radiosonde profiles from Munich (48.1° N, 11.6° E), Germany. The peak is more pronounced during winter months (DJF) compared to summer (JJA). Another important finding is the rather large discrepancy between the radiosonde data and the ECMWF ERA Interim reanalysis data set, which is largely due to the limited vertical resolution of the numerical model. Zhang et al. (2015) expand these results and present a ten year averaged annual cycle of tropopause-based vertical wind shear derived from radiosonde measurements taken at Boise, Idaho (43.6° N, 116.2° W). The results compare well with the ones from

Munich, exhibiting a more pronounced averaged vertical wind shear peak during winter, as well as on a larger vertical spread compared to summer. The meridional dependency of the occurrence of enhanced tropopause-based vertical wind shear was analysed by Zhang et al. (2019), based on 13 years of radiosonde measurements at 90 stations in the northern hemisphere. The high latitude stations are located in Alaska and show no strong signal in $S^2$ throughout the year. The extratropics are covered by stations in the US mainland, where the long year averaged data reveal a sharp $S^2$ maximum within the first kilometre

above the LRT between 20° N and 50° N. The tropics are represented by several stations in the Caribbean as well as in the western Pacific, and exhibit a more pronounced maximum in $S^2$ over a larger vertical distance above the LRT compared to the extratropics. The tropical tropopause-based $S^2$ maximum exhibits a bi-annual seasonality with maximum averaged values during DJF as well as during JJA.

The tropical summer maximum is associated with the emergence of the tropical easterly jet (TEJ) over the Indian Ocean,

which is an inherent component of the Asian summer monsoon circulation. Koteswaram (1958) was the first to describe these upper tropospheric easterlies, on the basis of radiosonde measurements at several stations between the equator and 40° N and from 20° W to 150° E, and he already noted exceptionally pronounced vertical wind shear above the wind speed maximum in individual profiles. Sunilkumar et al. (2015) investigated occurrence frequencies for vertical wind shear threshold values in the context of turbulence characteristics over India, based on a 4 year data set of GPS-radiosonde measurements at Trivandrum

(8.3° N, 76.6° E) and Gadanki (13.5° N, 79.2° E). The analysis reveals a sharp peak of maximum occurrence frequencies for enhanced vertical wind shear close to the cold point tropopause (CPT) and particularly during summer, even though the authors did not make use of a tropopause-based vertical coordinate.

The occurrence of exceptional vertical wind shear in the tropopause region can exceed the static stability and result in subcritical Richardson numbers, the emergence of dynamic instabilities, wave overturning and turbulent breakdown of the

flow. This was the central subject of a variety of observational studies in the context of clear air turbulence (CAT) and STE (e.g. Shapiro, 1976, 1978). Recently, Kunkel et al. (2019) analysed the relation of the TIL and STE (e.g. Gettelman and Wang, 2015) based on a synergistic approach analysing in situ airborne measurements from the WISE field campaign along with ECMWF Integrated Forecast System (IFS) data and idealised numerical baroclinic life cycle experiments. They could show that enhanced wind shear emerged in regions of enhanced static stability above the local tropopause, i.e., the TIL. In





their case this resulted in subcritical Richardson numbers, turbulent mixing and STE. Kaluza et al. (2019) describe a general co-occurrence of enhanced tropopause-based static stability $N^2$ and enhanced tropopause-based vertical wind shear $S^2$ in breaking baroclinic waves as a generic feature in midlatitudes, based on a composite analysis of operational ECMWF IFS analysis data. The composites show overall reduced Richardson numbers within the lowermost stratosphere (LMS, Holton, 1995) in ridge regions of breaking baroclinic waves. These results suggest that enhanced wind shear is evident close above the

local tropopause, at least, in the extratropics, and might lead to the generation of turbulence and subsequent STE.

The previous paragraphs gave an overview of observational evidence of enhanced tropopause-based vertical wind shear. A majority of the shear regions described are associated with planetary circulation features, i.e. the polar jet in midlatitudes, the subtropical jet stream (STJ) over east Asia and the northwest Pacific, and the summer TEJ over the Indian Ocean. The following paragraphs recapitulate processes that influence the occurrence of wind shear, particularly in the upper troposphere/lower

stratosphere (UTLS) region. The jet streams present the planetary-scale background state for the distribution of wind shear, with vertical shear zones spanning over several kilometres and lateral shear zones of several hundreds of kilometres. The polar or eddy-driven jet is associated with baroclinic development at midlatitudes, and can be described in the first approximation as a thermal wind. The upper level front above the level of maximum horizontal wind speed is associated with exceptionally large horizontal temperature gradients, which in turn causes enhanced vertical wind shear according to the thermal wind relation. The

analysis of Endlich and McLean (1965) showed a general agreement between the thermal wind which was calculated based on temperature measurements and a vertically confined layer of enhanced vertical wind shear derived from wind measurements.

The subtropical jet stream (STJ) is located closer to the equator and generally at higher altitudes compared to the polar jet, and it is driven by the conservation of angular momentum during the poleward excursion of air masses, as well as the temperature gradient at the subtropical tropopause break. The winter STJ over east Asia and the west Pacific exhibits exceptionally high

wind speeds (Jaeger and Sprenger, 2007) as well as large occurrence frequencies (Koch et al., 2006), and is also referred to as the east Asian jet stream (EAJS, Yang et al., 2002). This fact, however, is not reflected in the occurrence frequency of reduced Richardson numbers or enhanced turbulence index ($TI$) values at tropopause altitudes (Jaeger and Sprenger, 2007), which might indicate that the tropopause-based wind shear is not as pronounced compared to other jet streams. It should, however, be considered that neither the Richardson number nor the $TI$ are solely defined by $S^2$. The occurrence of a tropopause-based

vertical wind shear peak, as it is apparent in the observational studies, is in fact not necessarily linked to exceptionally large wind speed, because of the limited vertical extent of the shear regions as well as due to the fact that directional shear can contribute to the total wind shear. The summer TEJ presents a descriptive example for this issue. The upper-tropospheric easterlies which define the TEJ exhibit average wind speeds around 40 m/s, which is rather slow compared to the winter STJ and polar jet. They are however associated with the most pronounced tropopause-based maximum in $S^2$ (Sunilkumar et al.,

2015; Zhang et al., 2019).

The planetary scale background wind shear which is determined by the occurrence of the jet streams is further modulated by smaller scale processes. Liu (2017) confirm the relevance of processes on a large spectrum of scales through spectral decomposition of the tropopause-based wind shear features in numerical model data from the NCAR Whole Atmosphere Community Climate Model (WACCM). On the mesoscale, flow deformation, convergence and differential temperature advection can result





in frontal zones with the associated wind shear according to the thermal wind relation (Ellrod and Knapp, 1992). Kunkel et al. (2014) showed in numerical baroclinic life cycle simulations, how mesoscale gravity waves with intrinsic frequencies close to the inertial limit (inertia gravity waves, IGW) can deform the flow, resulting in local enhancement of $S^2$ in the tropopause region. Gravity waves furthermore interact with the tropopause in general and the TIL in particular (Kunkel et al., 2014; Zhang et al., 2015, 2019), since the maximum in $N^2$ presents a maximum in the refractive index, resulting in wave refraction and a

characteristic shift in the frequency spectrum, as well as partial or total wave reflection.

Bense (2019) found the transmission coefficient and the shift of the wave spectrum to exhibit a rather complex dependency on the wavelength, the angle of the wave vector, the strength and depths of the TIL, as well as the background wind profile. A case study in this work furthermore showed how critical level filtering in the tropopause region of orography-induced gravity waves resulted in a pronounced local wind shear enhancement caused by non-linear wave-mean-flow-interaction. The

overall relevance of critical level filtering at the tropopause, however, depends on the wave spectrum and the background flow, and remains to be quantified. For example, Spreitzer et al. (2019) analysed the diabatic tendencies of the physical process parametrisations in the ECMWF IFS model during a baroclinic life cycle case study. The tropopause region within the ridge of the baroclinic wave featured regions of pronounced $S^2$ enhancement, reduced Richardson numbers, and diabatic PV modification. The contribution of the non-resolved gravity wave drag parametrisation, however, was found to be negligible.

The interrelation between gravity waves, enhanced vertical wind shear and the TIL is particularly of interest because of their common region of occurrence. Ridges of baroclinic waves are often associated with strongly curved jet streaks, jet exit regions, deep convection, and frontal zones. These features are sources for the emission of a large intrinsic frequency spectrum of gravity waves, reaching from the inertial limit up to small scale high frequency waves (Plougonven and Zhang, 2014). These waves propagate upwards towards the tropopause within ridges, which are associated with pronounced TIL enhancement as

well as a frequent occurrence of enhanced vertical wind shear (e.g. Kaluza et al., 2019). IGWs can sharpen the synoptic scale background wind shear associated with the jet stream locally on the mesoscale, potentially resulting in subcritical Richardson numbers, with eventual growth of a small scale perturbations due to dynamic instabilities (Sharman et al., 2012). The evolution of dynamic instabilities, wave growth and overturning, and eventual turbulent breakdown results in the erosion of the gradients responsible for the initial instability, as well as gradient sharpening in the vertically adjacent regions, which

can lead to the vertical succession of momentum gradients (Reiter, 1969).

Fritts et al. (2013) and subsequent research by the lead author (Fritts et al., 2016, 2017) analyse the *sheet and layer* structure of the atmosphere, a feature which is closely linked to the occurrence of enhanced tropopause-based vertical wind shear. They use a direct numerical simulation (DNS) setup which consists of a large scale vertically propagating gravity wave in a background flow with a superimposed fine-scale structure, defined by a static small-scale wave perturbation. The phase

relation between the background and the gravity wave results in regions of turbulence and gradient erosion, with adjacent gradient sharpening, resulting in layers of reduced gradients, and thin sheets of enhanced gradients up to a factor 10 of the background values.

The preceding paragraphs gave an overview of observational and theoretical research studies on the occurrence of enhanced vertical wind shear $S^2$ in the tropopause region. To the knowledge of the authors, a comprehensive statistical analysis of this





atmospheric feature does not yet exist, particularly on climatological scales. This work presents an approach towards such
an analysis, on the basis of ten years of northern hemispheric ECMWF ERA5 reanalysis data. The ERA5 reanalysis presents
a consistent, state-of-the-art long-term representation of the atmosphere, with a sufficient resolution to realistically resolve
central features in the UTLS, particularly gradient-based measures like static stability $N^2$ and vertical wind shear $S^2$. The
analysis addresses the following central questions: Where and how frequently is the tropopause region exposed to exceptionally
enhanced vertical wind shear, and how do these shear regions compare to atmospheric flow features and mechanisms that
modify momentum profiles?

The paper is structured as follows. In Sect. 2 we describe the data set as well as the data processing and general analysis
methods. In Sect. 3 we present an exemplary analysis a single day to present the issue and explain the metrics, and then
proceed with the 10 year climatology in Sect. 4. Sections 5 discusses the results, and Sect. 6 summarises the mayor outcomes
and presents a further outlook.

## 2   Data and methods

For this study we use ten years of daily northern hemispheric ECMWF ERA5 reanalysis fields (Hersbach et al., 2020), from 1
January 2008 to 31 December 2017, at 00:00 UTC. The reanalysis dataset is based on the Cy41r2 cycle of the ECMWF IFS
model, and for this study the output is processed on a regular $0.25°$ latitude-longitude grid, as well as on the 137 native vertical
hybrid sigma-pressure levels between the surface pressure and 1 Pa. This corresponds to an average vertical grid spacing in the
UTLS region of about 300–400 m, depending on the elevation of the tropopause. The region analysed is restricted from level
37 (counting towards the surface, with level 37 corresponding to about 25 km altitude) down to 1.5 km above the orography.
Thus, the planetary boundary layer (PBL) is excluded, as well as regions of enhanced wind shear in the mesosphere/lower
thermosphere (MLT) region (Liu, 2017).

Basic variables such as the temperature $T$ and the three-dimensional wind $(u, v, w)$ are directly provided by the ECMWF.
Based on the temperature profiles, the LRT altitude is determined following the definition of the WMO as the first level where
the lapse rate falls below 2.0 K km$^{-1}$ with the condition that it remains below this value between this level and all higher levels
within 2 kilometre above. The ten year analysis of tropopause-based wind shear at midlatitudes in Sect. 4.1 furthermore makes
use of dynamic tropopause fields provided by the ECMWF. Following the definition of Ertel (1942) the potential vorticity (PV)
can be written as

$$Q = \frac{1}{\rho} \boldsymbol{\eta} \cdot \times \boldsymbol{\nabla} \Theta \tag{1}$$

where $\rho$ is the density of the medium, $\boldsymbol{\eta} = \boldsymbol{\nabla} \times \boldsymbol{u} + 2\boldsymbol{\Omega}$ the vector of the absolute vorticity. The unit for the PV is the 'potential
vorticity unit' (pvu), with $1 \, \text{pvu} = 1 \cdot 10^{-6} \, \text{m}^2 \, \text{s}^{-1} \, \text{K} \, \text{kg}^{-1}$. The tropopause region is characterised by pronounced isentropic
gradients of the potential vorticity, and the dynamic tropopause in the extratropics is commonly defined by a constant value of
$Q = 2 \, \text{pvu}$ (Hoskins et al., 1985).

The vertical wind shear $S^2$ is calculated on half levels of the native vertical hybrid sigma-pressure level of the IFS, to retain
a maximum amount of information in the gradient based measure. The increasing vertical grid spacing with increasing altitude



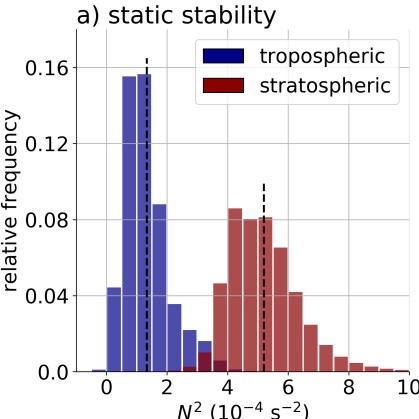
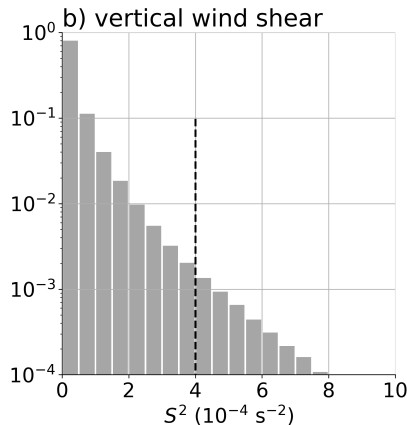

**Figure 1.** Relative occurrence frequency histogram of static stability $N^2$ and vertical wind shear $S^2$ between 1.5 km above the surface up to 25 km altitude, on 11.09.2017 at 00:00 UTC and over the northern hemisphere. Counts are weighted with the grid box volumes. a) Static stability ($N^2$, in $10^{-4}$ s$^{-2}$). Blue indicates tropospheric volume ($z < z(\text{LRT})$), red stratospheric volume ($z < z(\text{LRT})$). Black dashed lines indicate average tropospheric/stratospheric $N^2$. b) Relative occurrence frequency distribution of vertical wind shear ($S^2$, in $10^{-4}$ s$^{-2}$). Logarithmic occurrence frequency scale. Black dashed line indicates threshold value $S_t^2$.

in the native coordinates results in a bias towards a larger resolved spectrum of vertical wind shear at lower altitudes, which should be considered. The central goal of this study is to quantify the occurrence of enhanced vertical wind shear $S^2$. For this,

we define a threshold value $S_t^2 = 4 \cdot 10^{-4}$ s$^{-2}$, marking the top end of the spectrum of atmospheric vertical wind shear. The threshold value is selected based on the consideration that $S^2 \geq S_t^2$ generally can not be sustained by the average tropospheric static stability $N^2$, thus leading to low Richardson numbers and conditions favourable for turbulence. In contrast, average stratospheric static stability values of $S_t^2 = 4 \cdot 10^{-4}$ s$^{-2}$ lead to Richardson numbers of the order of magnitude of $\mathcal{O}(1)$. Figure 1 illustrates the previous consideration, for one exemplary time step on which we will also focus in Sect. 3. The distribution

of tropospheric static stability peaks around $N^2 = 1 \cdot 10^{-4}$ s$^{-2}$, with an overall average of $\overline{N^2}_{trop.} = 1.33 \cdot 10^{-4}$ s$^{-2}$. The stratospheric static stability distribution amounts to an average value of $\overline{N^2}_{strat.} = 5.20 \cdot 10^{-4}$ s$^{-2}$, which is shifted towards a larger value due to the above-average static stability which defines the TIL. The average stratospheric static stability above the TIL is generally closer to the value of $\overline{N^2}_{strat.} = 4 \cdot 10^{-4}$ s$^{-2}$ (e.g. Birner et al., 2002). The choice of the threshold $S_t^2$ is furthermore motivated by previous research studies which indicate the majorly exclusive occurrence of vertical wind shear

exceeding $S_t^2$ at tropopause altitudes (e.g. Kunkel et al., 2019).

## 3    Identification of a shear layer on 11 September, 2017

We will start our analysis by initially focussing on a specific time step in September 2017. Our intention is to introduce our metrics first and then look at longer time periods afterwards. We start with the synoptic situation to provide the large scale



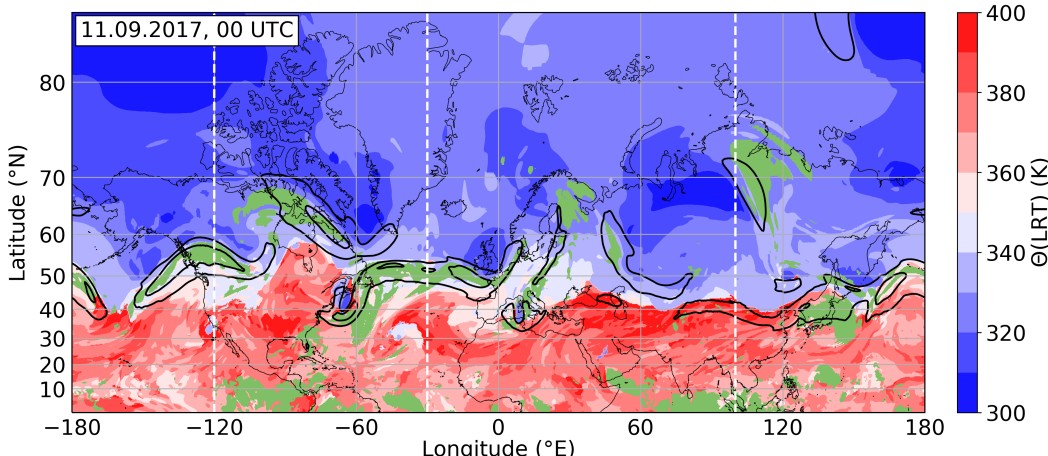

**Figure 2.** Potential temperature at the LRT ($\Theta(\mathrm{LRT})$, in K) over the northern hemisphere, on 11 September 2017 at 00:00 UTC. Black lines show contours of the vertically integrated horizontal wind, beginning at $30\ \mathrm{m\,s^{-1}}$ in steps of $15\ \mathrm{m\,s^{-1}}$. Green shaded areas indicate regions where $S^2$ exceeds $S_t^2$ within one km vertical distance from the LRT. Dashed white lines indicate the location of the vertical cross-sections in Fig. 3.

overview. Figure 2 shows a snapshot of the northern hemispheric potential temperature at LRT altitude on 11 September,

2017, along with maxima of the vertically integrated horizontal wind (Koch et al., 2006). The primary feature standing out is the tropopause break and the associated jet streaks of the horizontal wind. Over the Asian continent and reaching to the western Pacific, the tropopause break features a sharp meridional $\Theta(\mathrm{LRT})$ gradient with a single coherent STJ. Further west, the tropopause break is less sharp, and features a characteristic sequence of Rossby wave patterns accompanied by individual jet streaks at varying latitudes.

The synoptic scale wind systems are further illustrated in the vertical cross-sections in Fig. 3. At $120°$ W (Fig. 3a), a pronounced polar jet maximum is visible at $50-60°$ N, which is also evident in the vertically integrated horizontal wind in Fig. 2. Further southward, the tropopause break at $30°$ N features comparatively weak westerly winds, followed by high altitude easterlies in the tropics. The Atlantic (Fig. 3b) is dominated by a single pronounced jet streak located at midlatitudes. The cross-section at $100°$ E (Fig. 3c) again shows a sequence of zonal winds, with high altitude easterlies in the tropics,

followed by the STJ, and further northward the rotational components of a cyclonic system over Siberia.

We proceed with the analysis of the vertical and geographical distribution of strong vertical wind shear exceeding the threshold value $S_t^2 = 4 \cdot 10^{-4}\ \mathrm{s^{-2}}$. Figure 4 shows the zonally averaged relative occurrence frequency of strong vertical wind shear $S^2 \geq S_t^2$ on a logarithmic color-scale between $0.1\ \%$ and $10\ \%$, for the northern hemisphere on 11 September, 2017. Figure 4a reveals three meridional regions of exceptional occurrence frequencies located in the UTLS region, one in the tropics, one

at $30-40°$ N, and a third one at midlatitudes. Rearranging the grid volumes in a tropopause-based vertical coordinate system depending on their vertical distance from the LRT (Fig. 4b and c) concentrates the maxima of strong wind shear occurrence in





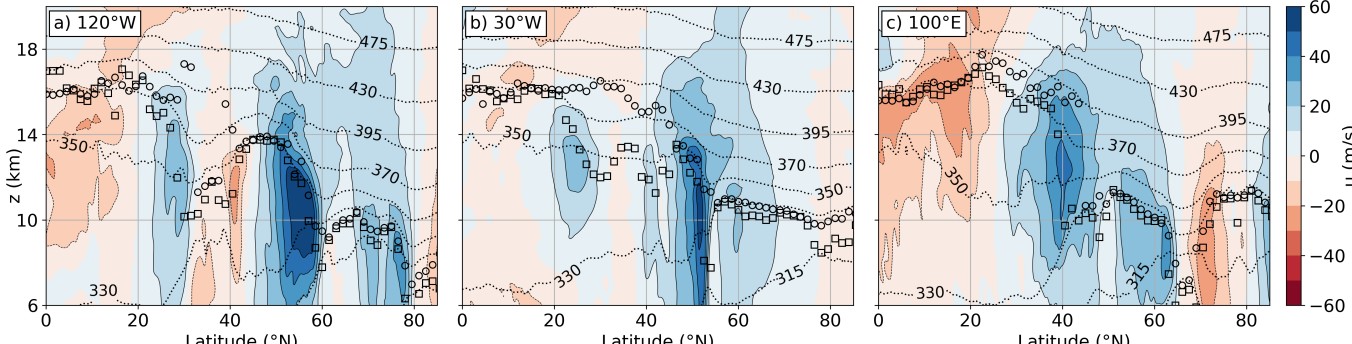

**Figure 3.** Vertical cross-sections on 11 September, 2017, at 00 UTC and at a) $120°$ W, b) $30°$ W, and c) $100°$ E. Colour contour shows zonal wind speed ($u$, in m s$^{-1}$), black dots LRT altitude, black squares $Q = 2$ pvu dynamic tropopause altitude, and black dotted lines isentropes ($\Theta$, in K).

all three regions to a vertical layer of about 1–2 km extent directly above the LRT. Secondary occurrence frequency maxima can be identified closely below the tropopause in all three meridional regions.

The pronounced occurrence of $S^2 \geq S_t^2$ in a distinct layer close to the LRT motivates to proceed with the analysis focussing
on the question *"Is there a global relation between vertical wind shear and the tropopause?"*. To answer this question, we choose a binary criterion where we define the tropopause region as being exposed to enhanced vertical wind shear if $S^2$ exceeds $S_t^2$ at least in one grid box volume within 1 kilometre vertical distance from the LRT. These regions are indicated in Fig. 2. The wind systems described in the first paragraph of this section can now be associated with the three meridional regions of enhanced tropopause based wind shear occurrence in Fig. 4. At midlatitudes, the tropopause region is exposed
to strong vertical wind shear within the Rossby wave jet streaks, as well as in the cyclonically curled up ridges associated with breaking baroclinic waves (over Canada, Scandinavia and Siberia) (Kaluza et al., 2019). Further southward, the STJ also features enhanced vertical wind shear at the tropopause, although not area-wide, and not correlated with the maximum of the vertically integrated wind. The tropical easterlies over the Indian Ocean and the east Pacific feature large scale coherent regions of enhanced vertical wind shear close to the tropopause. Several smaller scale tropopause based wind shear regions can
be attributed to individual atmospheric wind systems, e.g. the late stage ex-hurricane Irma which is located over Florida, or the stratospheric cut-off over the Mediterranean Sea.

## 4 Ten year climatology of enhanced wind shear in the northern hemisphere UTLS

### 4.1 Vertical distribution and seasonality of enhanced vertical wind shear in the northern hemisphere

We repeat the analysis steps from Sect. 3 for the whole dataset of ten years of daily ERA5 fields. We first present the overall
vertical distribution of enhanced vertical wind shear $S^2 \geq S_t^2$, followed by the geographic occurrence frequency and seasonality

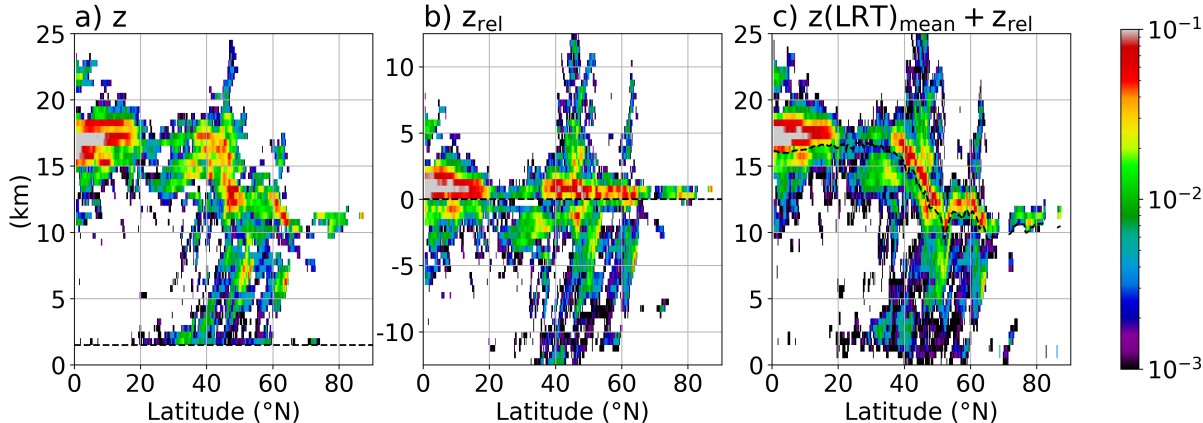

**Figure 4.** Relative occurrence frequency of $S^2 \geq S_t^2$, zonally averaged over the northern hemisphere, on 11 September, 2017 at 00 UTC. Logarithmic frequency contour, vertically binned in $dz = 500$ m. a) Absolute height distribution. Dashed black line indicates the effect of the 1.5 km above orography cut-off. b) LRT-based relative vertical coordinates. Dashed black line shows LRT altitude. c) As in b) but with the average LRT altitude of the shear regions restored.

in the tropopause region. Figure 5a shows the tropopause-based ten year temporal and zonal average occurrence frequency distribution for $S^2 \geq S_t^2$. The distribution compares well with the single day analysis. The northern hemisphere features a layer of occurrence frequency maxima of the order of 1 %–10 % within the first two kilometres above the LRT, spanning from the equator down to latitudes north of $60°$ N. In the following we will refer to the feature of maximum occurrence

frequencies at the LRT in the zonal average as a *tropopause shear layer (TSL)*, however, a comparison with the TIL should be made cautiously. Both features appear similarly in tropopause-based zonal means (compare e.g. Zhang et al., 2019), however, the wind shear layer emerges less frequently as well as less area-wide. Furthermore, a different metric is applied here, which analyses the tropopause-relative occurrence frequency of a threshold value $S_t^2$, instead of directly averaging $S^2$.

At midlatitudes, the TSL is composed of mostly decreasing wind speeds with height. Northward of about $45°$ N, the profiles

which exhibit $S^2 \geq S_t^2$ are on average associated with about 1 km more elevated LRT altitudes compared to the overall zonal average. This indicates the importance of large scale wave dynamics, particularly the polar jet streaks within ridges of baroclinic waves, which are known to exhibit exceptional wind shear in the tropopause region (Kaluza et al., 2019; Kunkel et al., 2019). The above-average LRT altitudes for profiles with $S^2 \geq S_t^2$ are a unique feature for the higher midlatitudes, as they do not occur in the tropics or the subtropics, which will be further addressed in Sect. 4.2.

At the tropopause break, the TSL is more evenly composed of both decreasing and increasing wind speed with height. This is due to the contribution of enhanced vertical wind shear at the upper edge of the tropopause break and above the jet core, as well as at the lower edge and below the jet core. The large vertical spread of the secondary occurrence frequency maxima above and below the tropopause break is at least partly due to enhanced vertical wind shear north of the upper edge of the





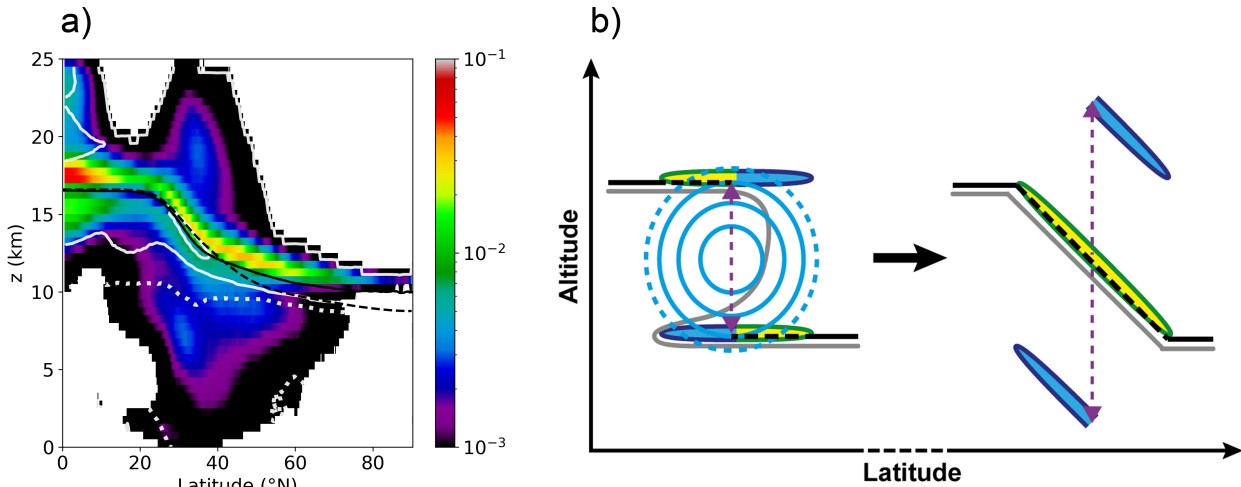

**Figure 5.** Northern hemispheric occurrence frequency distribution of grid volumes exhibiting $S^2 \geq S_t^2$, from January 01, 2008 to December 31, 2017. a) Vertical relative occurrence frequency distribution in the ten year tropopause-based zonal mean. Logarithmic frequency contour, vertically binned with $dz = 500$ m. Solid black line shows mean LRT altitude of profiles with $S^2 \geq S_t^2$, dashed black line overall averaged LRT altitude ($z(\mathrm{LRT})$, in km). White solid (dotted) line indicates regions where negative (positive) vertical wind shear makes for 75 % of the counts. b) Schematical vertical cross-section of the tropopause-based averaging method. Left part shows exemplary situation at the tropopause break, right part the key measures after tropopause-based averaging. Black lines indicate the LRT, grey lines the dynamic tropopause. Blue lines show isotachs, and the yellow and blue regions indicate regions of enhanced vertical wind shear above and below the jet core.

tropopause break, as well as within PV streamers below the jet core (Škerlak et al., 2015) that can sustain $S^2 \geq S_t^2$ but do not

meet the LRT definition. This issue is illustrated in Fig. 5b). The reader should keep in mind that the occurrence frequency distribution in Fig. 5a does not necessarily represent the extent of an instantaneous wind shear layer, but rather the general spread of enhanced wind shear occurrence around the tropopause.

The tropical region in Fig. 5a features the most pronounced occurrence frequency maximum, exhibiting a larger vertical spread compared to higher latitudes, as well as a distinct secondary maximum below the LRT. The composition of positive and

negative vertical wind shear in the tropics exhibits a layered structure, particularly above the LRT, where it is influenced by the quasi-biennial oscillation (QBO) phase and the elevation of the level of vanishing horizontal wind. We proceed with the analysis of the geographical spread of wind shear regions which are located within one kilometre vertical distance from the local LRT. For that, we calculate occurrence frequencies for how often the binary criterion for enhanced vertical wind shear close to the LRT which was defined in Sect. 3 is met.

Figure 6 shows the seasonality of these quasi-horizontal occurrence frequencies over the northern hemisphere. Throughout all seasons, a clear separation of occurrence frequency maxima is apparent, where each region can be attributed to a planetary circulation feature. The Northeast Pacific and the North Atlantic each exhibit a distinct maximum spanning over the storm track

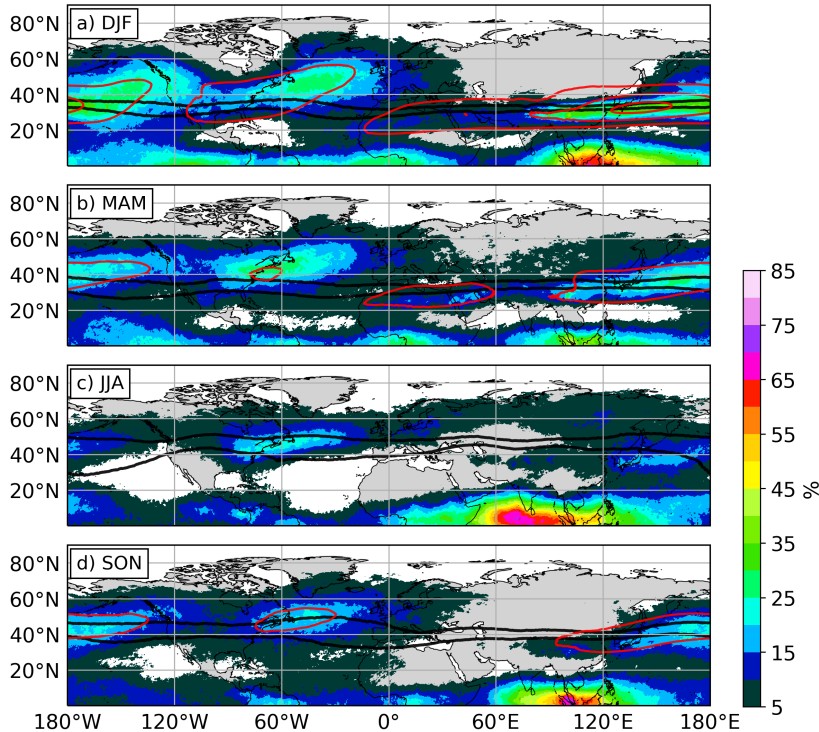

**Figure 6.** Occurrence frequency distribution of $S^2$ exceeding $S_t^2$ within 1 km vertical distance from the LRT and in the northern hemisphere. Averaged over 10 years from 2008 to 2017 for a) DJF, b) MAM, c) JJA and d) SON. Red solid lines indicate isolines of the vertically integrated wind, starting at 30 m s$^{-1}$ in steps of 15 m s$^{-1}$. Black solid lines show averaged location of the $\Theta(Q = 2\ \mathrm{pvu}) = 340$ K and $\Theta(Q = 2\ \mathrm{pvu}) = 360$ K isolines.

regions (Shaw et al., 2016). These maxima are most pronounced during winter and spring, exhibiting maximum occurrence frequencies of $25 - 30$ %.

Another region with maximum occurrence frequencies exceeding 30 % is apparent over Asia and the Northwest Pacific during winter, which is associated with the STJ. The strength of the maximum throughout the year shows a general agreement with the seasonality of the jet stream. The tropics feature several maxima spanning around the equator, with one particular wind shear region located over the Indian Ocean which exhibits maximum occurrence frequencies up to 70 % during JJA. This maximum shifts eastward during autumn and winter, where it is located centred over the maritime continent. The summer

maximum can be attributed to the emergence of the TEJ during these months associated with the Asian summer monsoon circulation. The winter maximum is linked to the Pacific Walker circulation and the El Niño Southern Oscillation (ENSO) ocean-atmosphere coupling. This will be further addressed in Sect. 4.3.





The following subsections each focus on one of the regions of frequent enhanced tropopause-based vertical wind shear occurrence, beginning at high latitudes and moving towards the equator. The goal is to narrow down the relation to planetary circulation features, and to investigate formation mechanisms of the TSL in the dataset.

### 4.2 The TSL related to Rossby waves in the mid latitudes

At midlatitudes, the occurrence of enhanced tropopause-based vertical wind shear is linked to the jet streaks of the polar or eddy-driven jet, and therefore to the associated barclinic wave patterns. Figure 5a indicated above-average LRT altitudes for profiles exposed to $S^2 \geq S_t^2$, which hints towards the role of the ridges of baroclinic waves. This behaviour is agrees with the conclusions from the process study of Kunkel et al. (2019) as well as the composite study of baroclinic waves of Kaluza et al. (2019). The following subsection presents a more detailed analysis of the dependency of enhanced vertical wind shear on the tropopause altitude.

We begin by selecting two zonal regions encompassing the occurrence frequency maxima in the midlatitudes, one over the North Atlantic from $80°$ W to $0°$ W, and a second one over the Northeast Pacific from $180°$ W to $120°$ W. The latter selection is made in such a way to exclude the STJ maximum to the west. We proceed with the $Q = 2$ pvu isosurface fields provided by the ECMWF, and make use of the conservation property of the potential temperature $\Theta$ on such an isosurface under adiabatic and frictionless conditions (Hoskins, 1991). We can therefore identify Rossby waves through anomalies of $\Theta(Q = 2\,\mathrm{pvu})$, with positive anomalies indicating a ridge of a baroclinic wave, i.e., subtropical air masses with high tropopause altitudes reaching poleward and negative anomalies indicating a trough, i.e., polar/subpolar air masses with low tropopause altitudes reaching equatorwards.

We proceed by defining a background state from which we can identify $\Theta(Q = 2\,\mathrm{pvu})$ anomalies. Figure 7a and b show the ten year temporal and zonal average of the potential temperature $\overline{\Theta}(Q = 2\,\mathrm{pvu})$, for both latitudinal regions and subdivided into DJF, MAM, JJA and SON. On average, the dynamic tropopause exhibits potential temperatures of about 380 K in the tropics, and decreases to values between $300 - 320$ K in the polar region. The meridional $\overline{\Theta}(Q = 2\,\mathrm{pvu})$ gradient is most pronounced during DJF and MAM in both regions. We define $\overline{\Theta}(Q = 2\,\mathrm{pvu})$ in the meridional region from $35°$ N–$60°$ N as the background state for the following analysis. According to Fig. 6, this region encompasses the occurrence frequency maxima over the North Atlantic and the Northeast Pacific. Furthermore, Fig. 7a and b show that the interseasonal standard deviation of $\overline{\Theta}(Q = 2\,\mathrm{pvu})$ is generally small in this region, which indicates that the ten year seasonal average describes representative average potential temperatures on the dynamic tropopause for the individual years. Figure 7c and d show for the zonal mean that the LRT exhibits averaged potential temperatures close to the ones of the dynamic tropopause, particularly in the region of interest from $35°$ N–$60°$ N. We present this comparison because we make use of both tropopause definitions. On one hand, we analyse the occurrence of $S^2 \geq S_t^2$ relative to the LRT, because $S^2$ is directly linked to the thermal stratification through the dynamic stability restriction. On the other hand, we identify UTLS wave features on the basis of the dynamic tropopause, because of the conservation property of the PV.

We proceed by calculating the distribution of instantaneous deviations from $\overline{\Theta}(Q = 2\,\mathrm{pvu})$ during the ten years, for each region and season. Figure 8 shows the result for the North Atlantic region, during DJF and at $51°$ N. The $\Theta(Q = 2\,\mathrm{pvu})$



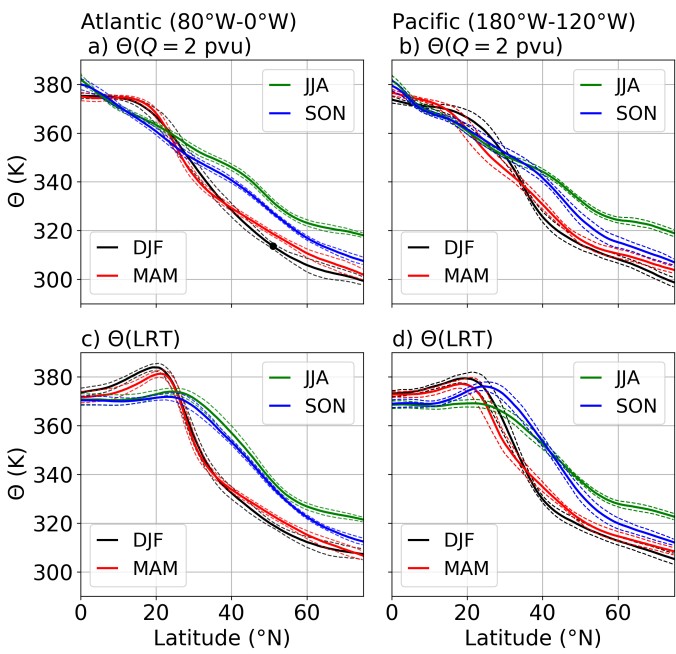

**Figure 7.** a) and b): Seasonal ten year temporally and zonally averaged dynamic tropopause potential temperature ($\overline{\Theta}(Q = 2\,\mathrm{pvu})$, in K). c) and d): Seasonal ten year temporally and zonally averaged LRT potential temperature ($\overline{\Theta}(\mathrm{LRT})$, in K). Solid lines show mean values, dashed lines interseasonal standard deviation. Left column North Atlantic region ($80°$ W–$0°$ W). Right column Northeast Pacific region ($180°$ W–$120°$ W).

values are unimodally distributed around the ten year average background value of $\overline{\Theta}(Q = 2\,\mathrm{pvu}) = 313.6$ K. Vertical wind shear exceeding $S_t^2$ in the vicinity of the LRT however occurs almost exclusively at above-average $\Theta(Q = 2\,\mathrm{pvu})$ values. The occurrence frequencies for $S^2 \geq S_t^2$ increase up to $50\,\%$ with increasing $\Theta(Q = 2\,\mathrm{pvu})$, up to the point where the occurrence

frequencies for the potential temperatures become statistically insignificant.

Figure 9 presents the extension of the previous analysis for both regions, all seasons, and over the whole selected meridional range. Figure 9a puts the result from Fig. 8 into a larger context, and reveals that $S^2 \geq S_t^2$ in the vicinity of the LRT is generally associated with positive $\Delta\Theta(Q = 2\,\mathrm{pvu})$ anomalies, i.e. dynamic tropopause potential temperatures above the climatological mean, i.e. primarily ridge regions. The occurrence frequencies increase with increasing potential temperatures and at higher

latitudes, with peak values exceeding $50\,\%$ at about $50°$ N and $\Delta\Theta(Q = 2\,\mathrm{pvu}) \approx 20$ K.

The general dependency for the occurrence frequency of $S^2 \geq S_t^2$ in the latitude-$\Delta\Theta(Q = 2\,\mathrm{pvu})$-coordinates holds true for both regions and all four seasons, with overall reduced values towards the summer months, accompanied by a northward shift. In the Northeast Pacific region and particularly during DJF, the influence of the STJ is visible, causing the bimodal distribution of $\Delta\Theta(Q = 2\,\mathrm{pvu})$ at low latitudes. This is due to occasional northeastward excursions of the STJ and the subtropical

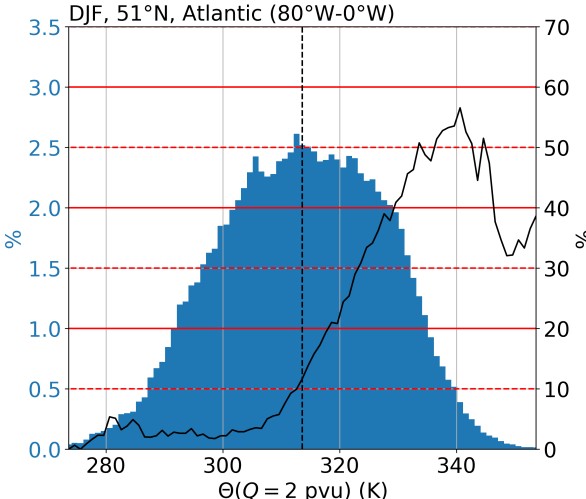

**Figure 8.** Occurrence frequency of $S^2 \geq S_t^2$ within 1 km vertical distance from the LRT, depending on $\Theta(Q = 2\,\mathrm{pvu})$. Blue histogram shows occurrence frequency distribution of $\Theta(Q = 2\,\mathrm{pvu})$ at $51°$ N during DJF 2008–2017 and averaged over $80°$ W–$0°$ W. $\Theta(Q = 2\,\mathrm{pvu})$ in 1 K bins. Dashed black line shows mean value $\overline{\Theta}(Q = 2\,\mathrm{pvu}) = 313.6$ K (black dot in Fig. 7a). Solid black line shows occurrence frequency for $S^2 \geq S_t^2$ in 1 km vertical distance from the LRT within the $\Theta(Q = 2\,\mathrm{pvu})$ bins.

tropopause break into the region of interest, which adds exceptionally high dynamic tropopause potential temperatures to the statistic and shifts the overall average towards larger $\overline{\Theta}(Q = 2\,\mathrm{pvu})$ values.

The following consideration is made to put the results of the previous paragraphs in context. From a static point of view and according to Fig. 6a, maximum occurrence frequencies for $S^2 \geq S_t^2$ in the vicinity of the LRT over the North Atlantic and during DJF are located at about $50°$ N and reach up to values of $25 - 30$ %. From a dynamic point of view, the occurrence can

be further narrowed down, with occurrence frequencies exceeding $45$ % within any elevated tropopause between $80°$ W–$0°$ W and $45°$ N–$55°$ N that exhibits $\Delta\Theta(Q = 2\,\mathrm{pvu}) \gtrsim 20$ K. The northward excursion of air masses exhibiting above average tropopause altitudes within ridges of baroclinic waves and the associated jet streaks can be expected to be mainly responsible for this correlation.

Overall, the analysis presents one way to quantify the link between the occurrence of tropopause-based enhanced vertical

wind shear and the potential temperature of the dynamic tropopause, which can be interpreted as a measure for the excursion of air masses within baroclinic waves in the midlatitudes.




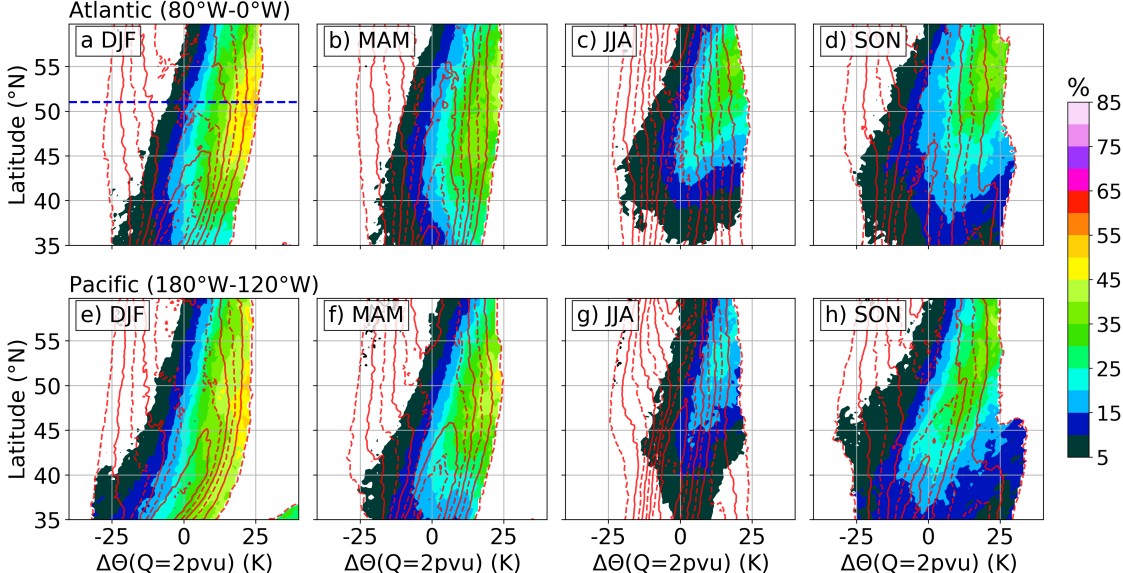

**Figure 9.** Occurrence frequency of $\Delta\Theta(Q = 2\,\mathrm{pvu})$ deviations from $\overline{\Theta}(Q = 2\,\mathrm{pvu})$ as in Fig. 7c, with meridional dependency. Dashed (solid) red lines indicate occurrence frequencies isolines, beginning at $0.5\,\%$ $(1.0\,\%)$ and in steps of $1.0\,\%$. Colour contour shows occurrence frequency for $S^2 \geq S_t^2$ in 1 km vertical distance from the LRT within the $\Theta(Q = 2\,\mathrm{pvu})$ bins. a) to d) DJF MAM JJA SON over the North Atlantic region. e) to h) Same for the Northeast Pacific region. Blue dashed line in a) indicates location of Fig. 7c

### 4.3 The TSL around the subtropical jet stream and in the tropics

#### 4.3.1 The east Asian jet stream

The tropopause region over east Asia and the northwest Pacific features a maximum in occurrence frequencies for $S^2 \geq S_t^2$,

which is associated with the STJ (Fig. 6). The STJ in this region is most pronounced during DFJ, and is also referred to as the east Asian jet stream (EAJS). Figure 10 shows the zonal and ten year seasonal average of the potential temperature $\Theta$ at (dynamic and lapse rate) tropopause altitudes for DJF and JJA, as well as the associated occurrence frequency of $S^2 \geq S_t^2$ within 1 kilometre vertical distance from the LRT. The meridional course of $\Theta(\mathrm{LRT})$ agrees well with the climatology derived from radiosonde measurements by Seidel et al. (2001). During winter, the zonally averaged occurrence frequency maximum

for $S^2 \geq S_t^2$ is located right at the meridional location of the tropopause break, and exhibits values of up to $25\,\%$ at around $30°$ N. The strength of the occurrence frequency maximum exhibits a pronounced interseasonal standard deviation of $5\,\%$ (absolute values), which can be partly linked to variability features of the EAJS. The meridional location of the occurrence frequency maximum is rather sharply defined, in agreement with the stable meridional location of the EAJS core on interannual time scales (Yang et al., 2002). The zonal location of the jet core however exhibits a comparatively large interannual variability

(Wu and Sun, 2017), which can be linked to the interseasonal variability of the occurrence frequency of $S^2 \geq S_t^2$ in Fig. 10a.



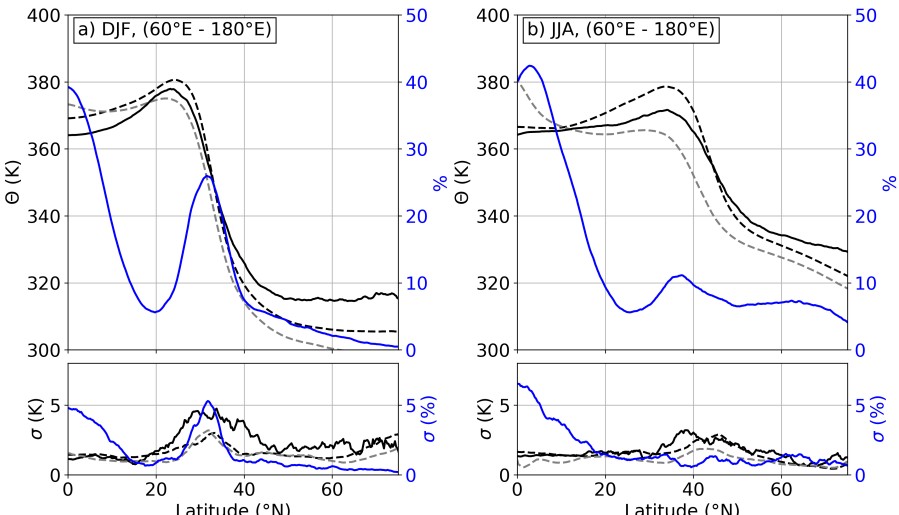

**Figure 10.** Vertical cross sections of ten year average tropopause properties between $60°$ E and $180°$ E longitude: a) winter (DJF) and b) summer (JJA). Upper panels: Dashed black line shows $\overline{\Theta}$(LRT) for the zonal region, dashed grey line $\overline{\Theta}(PV = 2 \text{ pvu})$. Blue solid line shows zonally averaged occurrence frequency for $S^2 \geq S_t^2$ within 1 km vertical distance from the LRT, and black solid line depicts $\overline{\Theta}$(LRT) for such vertical profiles. Bottom panels show interannual standard deviation for every measure. The standard deviation of the occurrence frequency displays absolute percentage values.

In the time period analysed by Wu and Sun (2017) that intersects with our analysis, they find the EAJS core to be located comparatively far westward during the winter seasons of 2008, 2011 and 2012, accompanied by large maximum EAJS core wind speeds. According to Fig. 11a, these years feature exceptionally large occurrence frequency maxima centred within the zonal region from $60°$ E–$180°$ E. In contrast, the 2009 and 2010 EAJS core is located further eastward and exhibits lower

maximum wind speeds, along with reduced occurrence frequencies for $S^2 \geq S_t^2$ (Fig. 11b). Note that the vertically integrated wind does not reflect the zonal location of the jet core. The link between the occurrence of tropopause-based enhanced vertical wind shear and the location and strength of the jet core should not be generalised and needs further investigation, since e.g. Wu and Sun (2017) do not see a strong correlation between the zonal EAJS core location and maximum wind speeds. Furthermore, it should be kept in mind that the occurrence of $S^2 \geq S_t^2$ in the vicinity of the LRT is not necessarily linked to exceptional jet

core wind speeds.

Towards summer, the jet stream slows down and shifts to the north of the upper-tropospheric Tibetan high pressure system associated with the East Asian summer monsoon (EASM) circulation. The zonally averaged occurrence frequency maximum for $S^2 \geq S_t^2$ in the vicinity of the LRT during JJA decreases to $11\,\%$, and is centred at the upper edge of the tropopause break (Fig. 10b).



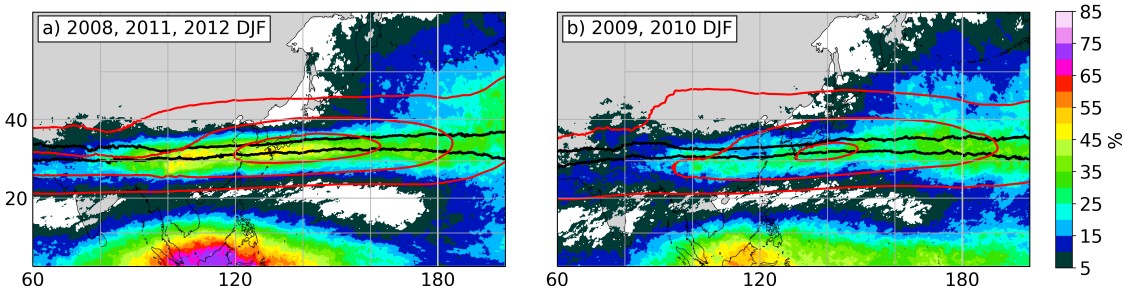

**Figure 11.** As in Fig. 6a for a smaller geographical section, averaged over selected DJF seasons of a) west-located EAJS core years (2008, 2011 and 2012), and b) normal located EAJS core years (2009 and 2010). For further destails see text and Wu and Sun (2017).

### 4.3.2 The tropical easterly jet

In the tropics, the summer maximum over the Indian Ocean (Fig. 6c) is associated with the tropical easterly jet (TEJ). These easterlies generally arise from June to September as part of the EASM circulation at upper-tropospheric pressure levels around 150 hPa (Krishnamurti and Bhalme, 1976). They do not exhibit a strong signal in the vertically integrated horizontal wind due to their limited vertical extent and comparatively low maximum wind speeds around 40 m/s. Vertical confinement and limited wind speed have an opposing effect on the vertical wind shear. Nevertheless, this results in the exceptionally large occurrence frequencies for $S^2 \geq S_t^2$ of up to 70 % in the vicinity of the the LRT over the Indian ocean (Fig. 6c). Figure 10b shows the zonally averaged occurrence frequency for $S^2 \geq S_t^2$ close to the LRT encompassing the TEJ summer maximum. It exhibits a significant interannual variability, a feature also seen in the vertical shear of the zonal wind found by Roja Raman et al. (2009) at different heights above the TEJ core using MST radar and GPS radiosonde measurements. Furthermore, the occurrence frequencies and their geographical distribution in Fig. 6c agree qualitatively well with the ones derived from radiosonde measurements by Sunilkumar et al. (2015). The authors analysed among others occurrence frequencies for vertical wind shear threshold values, however, not in a tropopause-based coordinate system, and with gradients calculated for $dz = 20$ meter (implicating a larger spectrum of resolved wind shears), which should be considered when comparing the results. The radiosonde data revealed occurrence frequencies during the monsoon season (June–September) for $S^2 > 2.25 \times 10^{-4}\,\mathrm{s}^{-2}$ of almost 80 % at Trivandrum (8.3° N, 76.6° E) and 67 % at Gadanki (13.5° N, 79.2° E), and for $S^2 > 9.0 \times 10^{-4}\,\mathrm{s}^{-2}$ occurrence frequencies of 37 % at Trivandrum and 15 % at Gadanki. These peak values occur as sharp maxima above the convective tropopause (COT) and close to the cold point tropopause (CPT), where the LRT is generally located in between (Sunilkumar et al., 2013).

### 4.3.3 The winter Pacific Walker circulation cell

During the winter months, the tropical tropopause region features another pronounced occurrence frequency maximum for $S^2 \geq S_t^2$ eastward of the summer TEJ maximum and centred over maritime continent (Fig. 6a). It is associated with localised



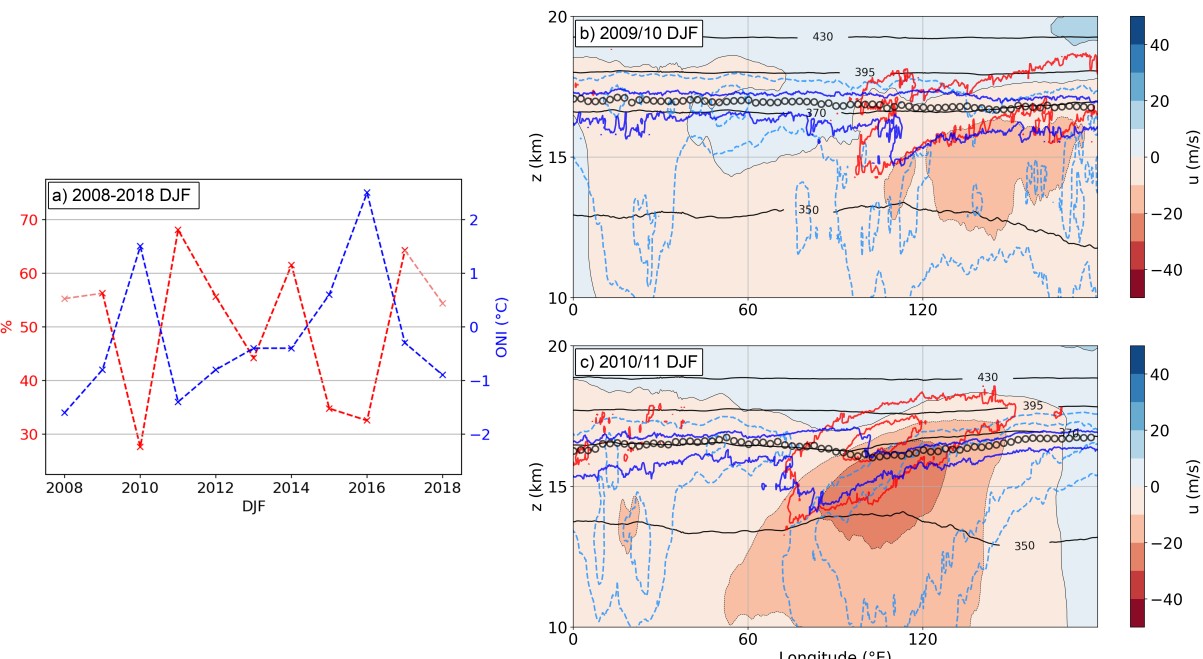

**Figure 12.** a) Comparison of the Oceanic Niño sea surface temperature anomaly Index for DJF (ONI, blue graph) with the strong shear occurrence frequency during DJF averaged over the region from $100°$ E to $130°$ E and from the equator to $10°$ N (red graph). The values for January–February 2008 and December 2017 are included (light red). ONI values from https://www.noaa.gov/. b) and c) Temporal mean of a zonal cross-section at the equator, averaged in geometric height coordinates, for the two consecutive DJF seasons 2009/10 and 2010/11. Color contour shows mean zonal wind component ($u$, in m s$^{-1}$). Red solid lines show the 10 % and the 50 % isolines of occurrence frequencies for $S^2 \geq S_t^2$. Blue solid lines show the 10 % and the 50 % isolines of occurrence frequencies for $N^2 \geq 4 \cdot 10^{-4}$ s$^{-2}$. Black circle markers indicate LRT altitude, and black solid lines show isolines of potential temperature ($\Theta$, in K). Light blue dashed lines indicate the isolines of 70 % and 85 % relative humidity over ice.

upper tropospheric easterlies, and exhibits a smaller zonal extent as well as about $5-10$ % lower maximum occurrence frequencies in the ten year average compared to the summer TEJ maximum.

The pronounced interannual variability in the zonally averaged occurrence frequencies for $S^2 \geq S_t^2$ (Fig. 10a), as well as the geographic location of the maximum and the season of its occurrence indicates a link to the El Niño Southern Oscillation (ENSO) ocean–atmosphere coupling and the effect on the Pacific Walker circulation cell. To test this hypothesis, we select a region encompassing the shear occurrence frequency maximum, from $100°$ E to $130°$ E and from the equator to $10°$ N, and calculate the average occurrence frequency of $S^2 \geq S_t^2$ close to the LRT in this region during DJF and for the individual years. The comparison of these time- and area-averaged frequencies with the Oceanic Niño sea surface temperature anomaly Index values for DJF (ONI) show an anticorrelation (Fig. 12a), with moderate to large occurrence frequencies during neutral






and La Niña conditions, and a peak during DJF 2010/11, a year of strong La Niña manifestation. The positive sea surface
temperature (SST) anomaly in the west Pacific is accompanied by increased convection over the maritime continent, which
results in enhanced upper tropospheric outflow and a strengthening of the adjacent Walker circulation cell (Fig. 12c). Still,
the frequent occurrence of enhanced tropopause-based vertical wind shear is striking, considering the still comparatively weak
easterlies responsible. The occurrence of enhanced tropopause-based vertical wind shear could be influenced by enhanced
gravity wave activity associated with deep convection (Podglajen et al., 2017), as convection is the major source for gravity
wave generation in the tropics through several forcing mechanisms (Müller et al. (2018) and therein). During the pronounced
El Niño winter seasons 2009/10 and 2015/16 the shear occurrence frequency reduces down to values in the range of 30 %,
which is 20 % below the 10 year average value of 50.4 %. This is associated with the eastward shift of the less sharply defined
rising branch of the Pacific Walker circulation cell (Sullivan et al., 2019), along with the less pronounced upper tropospheric
easterlies (Fig. 12b).

We now briefly revert to Figure 5a and the secondary occurrence frequency maximum for $S^2 \geq S_t^2$ below the tropical LRT,
which is strikingly separated from the one above. This secondary maximum emerges due to a comparable situation to the one
described for the secondary maxima at the subtropical tropopause break (Fig. 5b), only on smaller scales and zonally aligned.
The following explanation is applicable to both the TEJ as well as the the winter Walker cell easterlies. We will focus on
the latter to give an exemplary explanation. The easterlies responsible for the DJF TSL occurrence frequency maximum are
associated with a frequently occurring lapse rate tropopause jump of $0.5 - 1.0$ km vertical distance, located at the western
edge of the easterlies (Fig. 12c). This is indicated by the occurrence frequency isolines for $N^2 \geq 4 \cdot 10^{-4}$ s$^{-2}$. These regions
of enhanced static stability reach west- and downward, where at some point they do no longer fulfil the definition of the
LRT criterion, causing the tropopause jump. The tropopause-based averaging method then results in a secondary occurrence
frequency maximum for $S^2 \geq S_t^2$ below the LRT, due to enhanced wind shear within the regions of enhanced $N^2$ that reach
below the local LRT.

## 5    Discussion

The TSL and particularly its relation to the TIL owe their potential importance to the fact, that both the TSL and the TIL
majorly influence the dynamical stability and may be related to the formation of the extratropical transition layer (ExTL),
which is derived from observations of chemical tracers (e.g. Hoor et al., 2002; Pan et al., 2004). Related to that, the potential
implications of the frequently occurring TSL within the tropical tropopause layer (TTL, Fueglistaler et al., 2009) needs further
investigation.

It is important to analyse the relation of the TSL and the TIL in more detail, because both co-occur as climatological features
in zonal and/or temporal tropopause-based averages (e.g. Birner et al., 2002; Birner, 2006; Grise et al., 2010; Zhang et al.,
2019). However, we see no general co-location of enhanced $N^2$ and enhanced $S^2$ above the local lapse rate tropopause in
instantaneous considerations and on smaller scales. Ultimately this has consequences for the dynamic stability of the flow,
along with the potential for turbulence and mixing across and above the tropopause. In the case of baroclinic waves in the





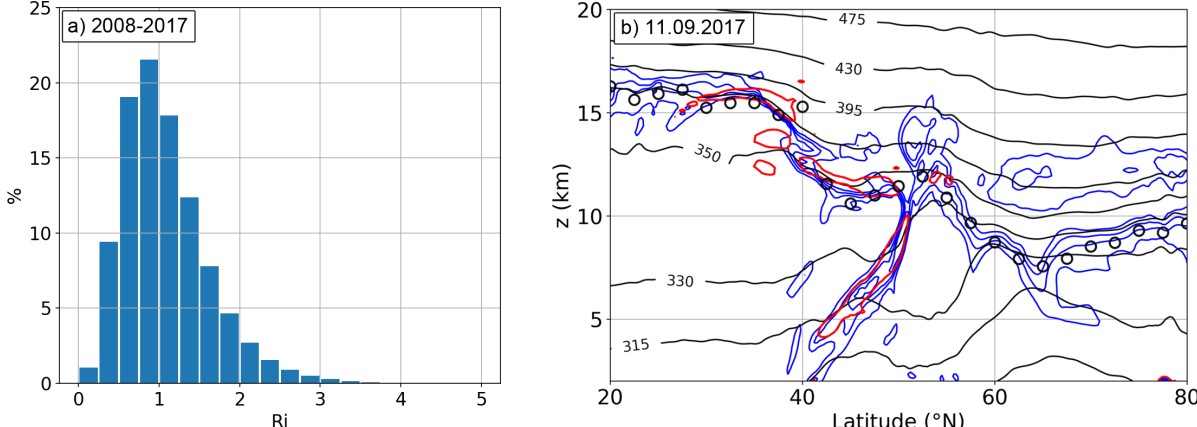

**Figure 13.** a) Histogram of the relative distribution of Richardson numbers associated with grid volumes that exceed $S_t^2$ within 3 km vertical distance above the local LRT. Data covering all daily fields from 2008–2017 and the whole northern hemisphere. b) Vertical cross-section at $60°$ W on 11 September 2017, at 00 UTC. Red lines show the $S^2 = S_t^2$ isoline, blue lines show PV at 2, 4, 6 and 8 pvu, and black lines indicate isentropes.

north Atlantic storm track region, Kaluza et al. (2019) describe a co-occurrence of the TIL and tropopause-based enhanced vertical wind shear in cyclone-based composites. This indicates that the processes responsible are active in the same region

on a synoptic scale, i.e. ridges. These processes however have a different impact on the temperature and momentum profiles, or even an exclusive direct effect on either. The vertical divergence of the vertical wind in the anticyclonic flow sharpens the temperature profile (Wirth, 2003), as well as long- and shortwave radiative effects due to trace gas and cloud ice water gradients at the tropopause (Randel et al., 2007; Kunkel et al., 2016), where the latter is particularly important in the context of warm conveyor belts and high reaching cloud formation in the tropospheric flow within ridges (Madonna et al., 2014). Gravity waves

can have a large influence on both $N^2$ and $S^2$ above the tropopause (e.g. Kunkel et al., 2014), however according to linear wave theory with a phase shift of $\pi/2$ between the perturbation of the horizontal wind $u'$ in direction of the horizontal wave vector $k$ and the perturbation of the temperature $T'$ (Suzuki et al., 2010), and a resulting shift between $N^2$ enhancement and $S^2$ enhancement (compare e.g. Kunkel et al., 2014). Overall, the relation between the TIL and the TSL needs further investigation, under consideration of the possibly varying importance of the forcing mechanisms at different latitudes.

Another closely related issue that needs further investigation is the relation between the TSL and the occurrence of turbulence and potential STE. Since most of the vertical wind shear that exceeds $S_t^2 = 4 \cdot 10^{-4} \text{ s}^{-2}$ occurs in a layer of 1–2 km above the tropopause, we expect the corresponding static stability to be of the same order of magnitude, according to the implications of the lapse rate tropopause definition for the stratification (Fig. 1). This is reflected in the histogram for the Richardson numbers of all grid volumes that exceed $S_t^2$ within 3 km above the LRT, during 2008–2017 and for the whole northern hemisphere (Fig.

13a). The Richardson number distribution peaks around $Ri = 1$, which agrees with the assumption we made concerning the




order of magnitude of $N^2$ compared to $S_t^2$. The variability of $Ri$ below and above this peak can be attributed to the variability of $S^2$ above the threshold value $S_t^2$, as well as to the variability of $N^2$ particularly in association with the TIL (Fig. 1).

We define the TSL as an occurrence frequency maximum for enhanced wind shear in tropopause-based zonal and temporal averages. This occurrence frequency maximum is composed of individual localised shear patches that emerge above the LRT,

as the exemplary cross-section in Fig. 13b shows, resulting in reduced Richardson numbers of the order of 1 in the elsewhere dynamically stable lower stratosphere. Enhanced wind shear does not correlate well with the occurrence of turbulence (Knox, 1997, e.g.) because it does not necessarily imply dynamic instability, but it is a prerequisite for dynamic instability to occur. Thus, the occurrence frequency distribution presented in Fig. 5a narrows down the region where turbulence due to dynamic shear instability can be expected above the tropopause. The vertical confinement within a few kilometres above the lapse

rate tropopause indicates a link to the chemically defined ExTL or mixing layer. Airborne in situ measured carbon monoxide (CO) profiles in the tropopause region exhibit a distinct "kink" in the vertical transition from tropospheric to stratospheric values, which defines the upper edge of a chemical transition layer. Hoor et al. (2004) determined it to be located at about 25 K potential temperature above the $Q = 2$ pvu dynamic tropopause, or 2–3 km above the lapse rate tropopause at high latitudes according to Pan et al. (2004). Hegglin et al. (2009) expanded these results onto a global scale based on CO-O$_3$ and

H$_2$O-O$_3$ tracer-tracer correlations derived from Atmospheric Chemistry Experiment Fourier Transform Spectrometer (ACE-FTS) satellite data. Berthet et al. (2007) used a statistical approach on Lagrangian trajectories that were subjected to TST, with their trajectory model being driven by operational ECMWF analysis data. The relative contribution of TST trajectories showed a largely tropopause-following behaviour with a limited vertical extent, indicating the amount and vertical reach of the permeability of the transport barrier that is the tropopause. Hoor et al. (2010) continued the Langrangian approach and linked

the chemical transition layer to troposphere-stratosphere transition transport timescales in the range of 0–50 days, which is short compared to the transport timescales in the LMS above the ExTL. The vertical confinement of the TSL may contribute to the separation of the LMS with a lower layer being frequently affected by shear driven turbulent STE (e.g Spreitzer et al., 2019; Kunkel et al., 2019). The TSL properties indicate its role as a contributing mechanism for the formation and maintenance of the ExTL, along with other known processes like e.g. radiation induced PV modifications (Zierl and Wirth, 1997) or convective

injection of tropospheric air (Homeyer et al., 2014).

## 6   Summary

The goal of this study was to investigate the occurrence of exceptionally pronounced vertical wind shear in the UTLS region. It was motivated by the wide range of research studies that describe the occurrence of exceptionally enhanced vertical wind shear at the tropopause, i.e. in the context of case studies on turbulent stratosphere-troposphere exchange (e.g. Shapiro, 1976, 1978;

Whiteway et al., 2004; Kunkel et al., 2019), in climatological studies based on radiosonde data and focussing on the TIL (Birner et al., 2002; Birner, 2006) and its interaction with gravity waves (Zhang et al., 2015, 2019), or in the context of planetary circulation features like the tropical easterly jet (Sunilkumar et al., 2015), and in numerical model data studies (Kunkel et al., 2014; Liu, 2017; Kaluza et al., 2019; Spreitzer et al., 2019). However, there was a lack of a comprehensive statistical analysis





of this atmospheric feature in a global and climatological sense. We approached this matter using ten years of data of daily
northern hemispheric ECMWF ERA5 reanalysis fields.

Exceptionally strong vertical wind shear was selected based on the threshold value $S_t^2 = 4 \cdot 10^{-4}\ \mathrm{s}^{-2}$. A single day analysis
introduced the metrics, and exemplified that enhanced vertical wind shear exceeding the threshold occurs nearly excusively in
close vertical vicinity above the lapse rate tropopause. It furthermore showed that the enhanced tropopause-based vertical wind
shear can generally be attributed to planetary circulation features, i.e. polar jet streaks within differently evolved baroclinic
waves, the subtropical jet stream, and the upper tropospheric easterlies of the Walker circulation cells.

The ten year temporally and zonally averaged meridional cross-section revealed a tropopause wind shear layer (TSL), which
is defined by a pronounced occurrence frequency maximum for enhanced tropopause-based vertical wind shear of the order of
$1\ \% - 10\ \%$. These occurrence frequencies are about an order of magnitude larger than anywhere else in the region analysed.
The TSL is vertically confined to the first 1–2 kilometres above the lapse rate tropopause, and spans from the tropics down to
high latitudes around $70°$ N. The geographical mapping of the TSL revealed distinctly separate regions of preferred occurrence,
that can each be linked to individual planetary circulation features:

- The storm track regions over the north Atlantic and the northeast Pacific feature distinct occurrence frequency maxima
  for enhanced tropopause-based vertical wind shear. These maxima are most pronounced during DJF, and are associated
  with jet streaks of the eddy-driven jet. The enhanced tropopause-based vertical wind shear emerges at above-average
tropopause altitudes associated with ridges of baroclinic waves. We identified the northward excursion of air masses
  within ridges on the basis of deviations of the potential temperature at the dynamic tropopause from the ten year temporal
  and zonal average within each region. The occurrence frequencies for enhanced tropopause-based vertical wind shear
  maximise at latitudes around $50°$ N, and reach values up to $50\ \%$ for profiles associated with above-average dynamic
  tropopause potential temperatures of $\Delta\Theta \approx 20$ K.

- The east Asian jet stream during winter exhibits occurrence frequencies for enhanced tropopause-based vertical wind
  shear of up to $30\ \%$, with a pronounced interannual variability. This appears to be linked to the variability in strength and
  location of the jet stream (Wu and Sun, 2017), which needs further investigation.

- The lapse rate tropopause region above the summer tropical easterly jet is exposed to enhanced vertical wind shear up
  to $70\ \%$ of the time and over a large area of the Indian Ocean. These very striking occurrence frequency maxima as well
as their interannual variability agree well with results from observation-based research studies (Roja Raman et al., 2009;
  Sunilkumar et al., 2015).

- The winter Walker circulation easterlies over the maritime continent exhibit an occurrence frequency maximum for
  enhanced tropopause-based vertical wind shear which is comparable to the one associated with the tropical easterly jet.
  The strength and zonal location of these easterlies is closely linked to the el Niño sea surface temperature anomaly and
the shift in the Walker circulation. The area-averaged occurrence frequencies for enhanced tropopause-based vertical





wind shear above the maritime continent show a pronounced interseasonal variability, ranging from 30 % during the el Niño phases in the winter seasons 2009/10 and 2015/2016, up to 70 % during the strong la Niña phase 2010/11.

Overall, this analysis presents a step towards a better understanding of the dynamic structure of the transition region between the troposphere and the stratosphere, i.e. the ExTL in the extratropics and the TTL in the tropics. The occurrence of the TSL can have several implications:

– The TSL could be an indicator for the efficiency of the processes responsible for its formation. The occurrence frequency for enhanced tropopause based vertical wind shear at the tropopause could for example correlate with the gravity wave activity, depending on the relevance of gravity waves for the formation of the TSL, i.e. their bi-directional interaction with the thermal stratification gradients that define the LRT and the TIL. The occurrence of the TSL in regions that are frequently exposed to enhanced gravity wave activity hints towards such a link, i.e. jet-front systems in baroclinic waves in the extratropics, and the rising branch of the Pacific Walker circulation.

– In this context, the occurrence of the TSL could also be an indicator for nonlinear wave-mean-flow-interaction and momentum deposition (Zhang et al., 2019; Bense, 2019). Enhanced vertical wind shear could be a residuum of preceding turbulence in vertically adjacent layers, as turbulent homogenisation can result in adjacent gradient sharpening.

– The TSL should have a noticeable intersection with the large scale forcing for tropopause regions that are susceptible to turbulent STE. This however needs thorough further investigation, as e.g. the differences between this analysis (Fig. 6) and the tropopause-based turbulence indicators by Jaeger and Sprenger (2007) show. The limitations of the numerical models used are an important factor to consider.

Overall, the work presented puts previous research studies on exceptionally pronounced vertical wind shear at tropopause altitudes into the context of the planetary circulation, and expands the results to present tropopause-based enhanced vertical wind shear as a global-scale UTLS feature. Conceptually, the top end values of the spectrum of atmospheric vertical wind shear in the UTLS are to be expected to occur in the tropopause region, because it fulfils the requirement of exceptional thermal wind gradient forcing as well as a stratification that can sustain the wind shear. The strict vertical confinement of the occurrence of $S^2 \geq 4 \cdot 10^{-4}\ \mathrm{s}^{-2}$ within the first two kilometres above the LRT however is quite striking, and it remains to be evaluated to what amount the wind-forcing and the required thermal stratification are mutually dependent. The role of additional processes like gravity wave propagation, refraction, and momentum deposition at the tropopause should be investigated in this context.

*Code and data availability.* ECMWF operational analysis data have been retrieved from the MARS server. Output from the data analysis steps and further information on the analysis code are available upon request (kaluzat@uni-mainz.de).

*Author contributions.* DK, PH, and TK designed the research project. TK developed the model code, performed the calculations, and analysed the data with the help of DK and PH. TK prepared the paper with contributions from all authors.





*Competing interests.* The authors declare that they have no conflict of interest.

*Acknowledgements.* The Authors acknowledge funding from the German Science Foundation, as this study was carried out in the context of the WISE campaign under funding from the HALO SPP 1294 (DFG grant no. KU 3524/1-1, HO 4225/7-1 as well as HO 4225/8-1). We are
560 furthermore grateful to the ECMWF for providing the ERA5 data which has been downloaded from the ECMWF MARS system.




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
