# Peer review of "On the occurrence of strong vertical wind shear in the tropopause region: A ten year ERA5 northern hemispheric study"

_Weather and Climate Dynamics, 2021_

## Author Response (AR1)

**Authors response**

We thank the anonymous referees for the interest in our study and the careful reading which helped to improve the paper. We hope that our replies to the comments answer the issues in a satisfying way, and that the changes in the manuscript motivated by the comments improved the paper. We first list the central changes in the revised version of the manuscript to give an overview (pages 1-2). Then we reply to the comments from the first referee (pages 2-13), followed by our response to the comments from the second referee (pages 14-24). We reply to each comment in a point-by-point response and in the following order: 1.) comment from the reviewer, 2.) our response and 3.) the changes in the manuscript. The modified text passages are given in italics.

**Central changes in the revised manuscript:**

– We changed the map projection in Fig. 2 from Mercator to Plate Carrée, to match the projection of Fig. 7 (respectively Fig. 6 in the previous version of the manuscript). We furthermore adjusted the colors in Fig. 2 to avoid the combination of red and green filled contours.

– Figure 3 in the revised manuscript now includes information on the occurrence of strong vertical wind shear $S^2 \geq S_t^2$.

– We identified an error in the plotting routine for the latitude-altitude crosssections which depict the temporal and zonally averaged occurrence frequencies for strong vertical wind shear. After a carefull revision of the plotting routine we recreated Fig. 4 and Fig. 5a (Fig. 5b in the revised manuscript). We show the changes here below (Fig. 5 and Fig. 6 in this reply). As can be seen, slightly larger values in the TSL occurrence frequencies are now apparent above the local tropopause. Overall, this revision has no effect on the interpretation of our results and the conclusion of our study.

– Figure 5 now includes an additional panel depicting the zonally averaged occurrence frequencies for strong vertical wind shear in absolute height coordinates (Fig. 5a). It furthermore includes a panel which depicts the zonally averaged occurrence frequencies relative to the cold point tropopause in the tropics respectively relative to the $Q = 2$ pvu dynamic tropopause in the extratropics (Fig. 5c). Both changes were motivated by comments from the two referees.

– The schematic of the averaging method (Fig. 5b in the previous version of the manuscript) is now included as an individual figure (Fig. 6 in the revised manuscript).

– Motivated by a comment from one of the referees we repeated each analysis step concerning the occurrence frequencies of strong vertical wind shear near to the tropopause with an adjusted vertical search range around the LRT to make the results more representative for both the tropics as well as the extratropics. The occurrence of strong vertical wind shear

near to the tropopause is now defined as $S^2 \geq S_t^2$ at least within one grid box between 1 km below and 2 km above the LRT. Thus, the occurrence frequencies in Fig. 7, 9, 10, 11, 12 and 13a in the revised manuscript are overall slightly larger (Fig. 6, 8, 9, 10, 11, 12a in the previous manuscript). This does not affect the interpretation of our results and the overall conclusion of this study.

– The horizontal map projections (Fig. 2, 7 and 12 in the revised manuscript) now depict the horizontal wind on the 200 hPa isobaric surface instead of the vertically integrated horizontal wind (Koch et al., 2006). After revising the manuscript we decided that the vertically integrated wind did not add significant information, and the more commonly depicted horizontal wind on an isobaric surface is more easily to interpret.

– We removed the information on the dynamic tropopause in Fig. 11 in the revised manuscript (Fig. 10 in the previous version) because it was not explicitly discussed in the text.

– Figure 13 in the previous version of the manuscript was replaced by Fig. 14 in the revised manuscript. Figure 14a shows a 2d-histogram of the distribution of $N^2$-$S^2$ pairs within the first three kilometers vertical distance from the LRT and for the whole ERA5 data set which is analysed in the study. Figure 14b shows the associated distribution of Richardson numbers in this region. These changes were motivated by comments from the referees.

**Point-by-point response to the first review**

**Main comments:**

**Comment 1:** The role of model vertical resolution is acknowledged in several instances but could be discussed more thoroughly. The authors mention in the introduction (p 3 l 67) that the previous generation of ECMWF reanalysis (ERA interim) was unable to describe the shear layers. How much improvement has ERA 5 brought ? The statement that it has 'a sufficient resolution to realistically resolve central features in the UTLS' (p 6 l 167-168) could be justified, although a comparison to radiosonde or other observations might be beyond the scope of this paper. The authors could refer to Figure 1 of Hoffman et al. (2019) which shows the respective resolution of ERA interim and ERA 5.Similarly, on p 6-7 l 197-199, the issue of vertical resolution variations with altitude is raised but its impact is not estimated quantitatively. The authors may want to mention that the change in vertical resolution is slight in the region of interest, a few percent (how many ?) compared to the changes in shear occurrence frequency which varies by 2 orders of magnitude in their Fig. 3.Finally, the authors could briefly comment on the performance of ERA5 compared to the ECMWF operational analysis which they analyzed in Kaluza et al. (2019).

**Reply to comment 1:** We realize that the additional information on the resolution of the ERA5 is necessary to put the results of the study into context, and have included the proposed changes into the manuscript.

*p.5 L160: The goal is to present a consistent area-wide analysis of the vertical and geographic occurrence frequency distribution for strong wind shear in a state of the art long term numerical representation of the atmosphere, i.e., the ERA5 reanalysis. Compared to observational research studies our approach has no spatial limitiations to assess the occurrence frequency. However, for our analysis it is necessary to keep in mind that the vertical wind shear features are only as well represented as the model resolution allows them to be. An important factor in this context is the vertical resolution, which has improved significantly compared to the ERA-Interim reanalysis as a reference dataset (e.g., Hoffmann et al., 2019).*

*p.7 L205: The increasing vertical grid spacing with increasing altitude in the native coordinates results in a bias towards a larger resolved spectrum of vertical wind shear at lower altitudes, which should be considered. However, the analysis focusses on the UTLS region where the vertical grid spacing increases from about 300 m at 5 km altitude up to about 400 m at 20 km altitude (e.g., Hoffmann et al., 2019). Thus, the resolution bias should not have a large impact on the results presented.*

*p.15 L352: We expect the ERA5 reanalysis to resolve vertical wind shear features in the UTLS similarly well compared to the operational IFS analysis data used in Kaluza et al. (2019). However, a direct comparison should be made carefully. The ERA5 reanalysis is based on an IFS version with a $T_L 639$ spectral truncation (about 31 km horizontal grid spacing, Hersbach et al. (2020)) and 137 vertical level, compared to the $T_L 1279$ spectral truncation (about 16 km horizontal grid spacing) and 91 respectively 137 vertical level in the operational analysis data used in Kaluza et al. (2019).*
* * *
**Comment 2:** Reading the paper made me wonder how much of the shear structure might be diagnosed/explained using the thermal wind relation (for instance the pattern in Fig. 4 and 5 a, the relationship with ridges in Sect. 4.2 or the seasonal variation of the EAJS in Sect. 4.3.1). The authors emphasize (p 24 l 547) that part of the co-location with the tropopause is related to thermal wind balance, as suggested for example in the cited study by Endlich and McLean (1965). They could test this hypothesis quantitatively. Sure, it does not directly explain how the temperature gradient are generated but this is documented elsewhere and a simple diagnosis could here help disentangle "balanced dynamics" from gravity wave effects.

**Reply to comment 2:** We thank the reviewer for this suggestion. We started to compare the wind shear based on the full model wind and the thermal (model) wind. We extended the discussion in Sect. 3 to include a comparison of these two metrics based on a single day analysis (Fig.15 in the manuscript, Fig. 1 in the this reply). The comparison shows that there is in general a good agreement in the larger scale in the extratropics, however, with distinct differences which are presumably mainly caused by resolved gravity waves. We have not included a climatological analysis of this comparison. We think that such a comparison would increase the content of this manuscript substantially. Also new metrics would need to be introduced, since the metrics used in this study are not suited for such a comparison. The zonal averaging of occurrence frequencies of $S^2 \geq S_t^2$

[Figure]

**Figure 1.** Comparison of the vertical wind shear based on the full model winds and on the thermal wind relation. a) Geostrophic wind at 200 hPa, for the northern hemisphere on 11 September 2017. Regions south of $20°$ N are left out because the validity of the assumption of geostrophic balance vanishes towards the equator. Black solid line indicates the vertical cross sections in panel b and c. b) Color contour shows vertical shear of the horizontal wind derived from the wind components in the ERA5 data ($S^2$, in ms$^{-2}$). Red and blue lines show $u = 30$ ms$^{-1}$ and $u = -10$ ms$^{-1}$ isotachs of the zonal wind. Black dotted lines show isentropes. Black circle markers indicate LRT altitude. c) As in b but for the thermal wind calculated from the temperature field in the ERA5 data. Red and blue lines show $u = 30$ ms$^{-1}$ and $u = -10$ ms$^{-1}$ isotachs of the zonal geostrophic wind.

as well as the quasi-horizontal mapping of strong wind shear near to the tropopause could show a good agreement between the full model wind shear and the thermal wind shear, even if the thermal wind does not represent the shear regions well, due to a superposition of underestimation and overestimation of the actual wind shear at different longitudes. This is indicated in the exemplary cross sections. We motivate further research on a more general comparison of the TSL with the thermal wind relation at the end of the paragraph.

*p.24 L514: The connection between the temperature field and the vertical wind shear for synoptic scale flow can be approximated through the thermal wind relation, i.e., the vertical gradient of the geostrophic wind under the assumption of hydrostatic balance. Figure 14a shows the geostrophic wind at 200 hPa on 11 September 2017, i.e., the date of the exemplary single day analysis in Sect. 3. Overall, the geostrophic wind approximates the synoptic scale flow realistically (compare Fig. 2). However, it overestimates the absolute zonal wind speed in cyclonic rotational systems like the one over the northwest Atlantic, in ac-*

*cordance with the fact that inertial forces are neglected (Holton and Hakim, 2012). The vertical structure of the geostrophic wind is shown exemplarily in the vertical cross sections at $60°$ W (Fig. 14b and c). The thermal wind relation results in several regions of strong vertical wind shear near to the tropopause. The comparsion with the vertical wind shear derived from the full model wind reveals a certain degree of agreement, in particular on the synoptic scale, but also differences on the smaller*

110 *scales. The strength of the vertical wind shear at $40°$ N and 15 km altitude is overestimated by the thermal wind approximation, as well as the shear region directly above that reaches north- and downward. The southward extent of the region of strong wind shear on the other hand is underestimated. The two pronounced wind shear regions below the tropopause and south of $40°$ N are not evident in the the thermal wind shear. The geostrophic zonal wind in the upper troposphere at about $50°$ N deviates from the full model zonal wind, which results in a significant underestimation the vertical wind shear below the tropopause.*

115 *At the same time, the thermal wind relation overestimates the shear region below the tropopause at $45°$ N, which is caused by strong meridional temperature gradients (not shown). The maximum of the thermal wind shear at $45°$ N directly above the LRT is not evident in the full model wind shear, but instead is apparent in a region that is located further to the north. Overall, the comparison indicates the significance of dynamic processes on smaller scales on which other forces than pressure gradient and Coriolis force need to be taken into account (Newton and Persson, 1962). This example already shows that many details,*

120 *especially related to mesoscale dynamic features, need to be considered to fully address the differences in the vertical wind shear based on the full model winds and on the thermal wind relation. A comprehensive analysis of these differences is beyond the scope of the current study but will be pursued in future work.*
* * *
125 **Comment 3:** If the resolution of ERA5 is good enough to distinguish the LRT from the cold point tropopause (CPT), it would be interesting that the authors determine which of the LRT or the CPT is closest to the enhanced shear layer in the tropics. This would be particularly relevant to the question of the stratosphere-troposphere boundary in the tropics (e.g. Pan et al., 2018) . I note that, in Fig. 5, the shear layer in the tropics is shifted upward by 1 pixel (500 m) with respect to the LRT.

130 **Reply to comment 3:** Motivated by this comment as well as a similar comment by the other reviewer, we have included the vertical distributon of grid volumes with $S^2 \geq S_t^2$ in the 10 year temporal and zonal average in a CPT-relative vertical coordinate for the tropics (Fig. 2c in this reply). It is possible to resolve the different locations of CPT and LRT in the tropics in ERA5. However, the occurrence of strong wind shear appears to be more closely linked to the stratification criterion that defines the LRT. The more pronounced occurrence frequency maximum below the CPT is closely linked to Fig. 13b and c and

135 the related discussion on the frequently occuring lapse rate tropopause jump within the TEJ and the winter easterlies over the maritime continent. The CPT is more often located above the downward sloping regions of strong wind shear.

*p.13 L307: The significance of the processes which result in the occurrence of the TSL remains to be quantified. The clustering of grid volumes which exhibit $S^2 \geq S_t^2$ directly above the LRT in Fig. 5b agrees with the dynamic stability criterion and*

140 *the thermal wind shear forcing associated with upper tropospheric fronts. However, the overall link between the tropopause*

[Figure]

**Figure 2.** Northern hemispheric occurrence frequency distribution of grid volumes that exhibit strong vertical wind shear $S^2 \geq S_t^2$, from 1 Januaray 2008 to 31 December 2017. Logarithmic frequency contour, vertically binned with $dz = 500$ m. a) Geometric altitude as the vertical coordinate. Solid bold black line indicates mean LRT altitude for all 10 years and the whole northern hemisphere. Dashed thin black line indicates the effect of the 1.5 km above orography cut-off. White solid (dotted) line indicates regions where negative (positive) vertical wind shear makes for 75 % of the counts. b) As in panel a, with LRT-relative vertical coordinate and with mean LRT altitude for profiles with $S^2 \geq S_t^2$ restored (dashed bold black line). Solid bold black line as in panel a. c) As in panel b but from $0° - 20°$ N with the cold point tropopause (CPT) as a reference altitude, and north of $20°$ N with the dynamic tropopause ($Q = 2$ pvu) as a reference altitude. Panels d to e: as in the top column but for $S_{t2}^2 = 6 \cdot 10^{-4}$ s$^{-2}$.

*definition, which is goverend by the temperature profile, and the occurrence of strong vertical wind shear remains uncertain.*
*Therefore, we repeat the analysis for the tropical cold point tropopause (CPT), i.e., the absolute temperature minimum in the*
*tropical UTLS, and for the dynamic tropopopause, i.e., the $Q = 2$ pvu isosurface in the extratropics. The tropics feature a*
*distinct separation of up to 1 km between the LRT and the CPT (Seidel et al., 2001), which motivates the comparison at low*

*latitudes. The PV on the other hand does not constitute a useful tropopause definition in the tropics (Holton, 1995), which is why the dynamic tropopause is only used in the extratropics in this study. From Fig. 5c it is evident that ERA5 resolves the separation of the LRT and the CPT in the tropics, along with central features like a decreasing mean distance between the two tropopauses towards the equator (Seidel et al., 2001). The clustering of strong wind shear grid volumes above the CPT is less pronounced compared to the LRT, along with a more pronounced secondary maximum below the CPT, which indicates that the occurrence of strong vertical wind shear is more closely linked to the LRT in the tropics.*

**Other comments:**

**Comment:** p 2 l 30: 'thermodynamic structure' : do you mean because of mixing and heat exchange? If yes, this should be explained. Otherwise, 'dynamic structure' or just 'structure' would fit better (wind shear is strictly speaking not a thermodynamic feature).

**Reply to comment:** We agree, thank you for the clarification. We changed the wording accordingly.

*p.2 L30: The distribution of vertical wind shear in the atmosphere is a substantial feature of the dynamic structure because it controls the dynamic stability of the flow.*

**Comment:** P 3 l 61: Please convert feet to meters, following WCD guidelines.

**Reply to comment:** Thank you for the hint.

*p.3 L60: The data showed distinct occurrence frequency maxima of enhanced values of $S^2$ within sampling windows of 0.9 km at altitudes of about 9-12 km.*

**Comment:** p 3 l 91: The authors may want to cite the recent paper by Trier et al. (2020). In this paper the occurrence of CAT around a mid-latitude cyclone is investigated with a special emphasis on its relation with gravity waves. This paper would also be relevant in the discussion, with the caveat that the small-scale waves are likely not resolved in ERA 5.

**Reply to comment:** We greatly appreciate this information, the study by Trier et al. (2020) is very informative and educational, and we agree that it should be referenced in our study.

*p.4 L97: Recently, the high resolution numeric simulation of a midlatitude cyclone which was associated with a large number of turbulence reports gave insight on the importance of the tropospheric jet streak and wind speed and shear enhancement within upper tropospheric outflow of deep convection on the occurrence of CAT, along with the generation of gravity waves on different scales and their interaction with the background wind shear profile at critical levels as well as in regions of subcritical Richardson numbers (Trier et al., 2020).*

**Comment:** p 6 line 191 : There is a typo in the definition of Q, with an extra × which should be removed (the dot is the conventional notation for the scalar product). Also, is it the 'full definition' which is used here, with **Ω** the vector of angular rotation? Although I imagine the differences will be small, I believe it should be replaced by $f\mathbf{k}$ where $\mathbf{k}$ for consistency with the primitive equations solved in the ECMWF model.

**Reply to comment:** We have corrected the typo in the equation, and we have added the definition of the potential vorticity as a conservation property in the primitive equations.

*p.6 L191: Following the definition of Ertel (1942) the potential vorticity (PV) can be written as*

$$Q = \frac{1}{\rho} \boldsymbol{\eta} \cdot \boldsymbol{\nabla}\Theta, \tag{1}$$

*where $\rho$ is the density of the medium, and $\boldsymbol{\eta} = \boldsymbol{\nabla} \times \boldsymbol{u} + 2\boldsymbol{\Omega}$ the vector of the absolute vorticity with the angular velocity of the earth $\boldsymbol{\Omega}$. In the context of the primitive equations which are solved by the IFS this translates to*

$$Q = \frac{1}{\rho}(f\boldsymbol{k} + \boldsymbol{\nabla} \times \boldsymbol{u_h}) \cdot \boldsymbol{\nabla}\Theta, \tag{2}$$

*where $f = 2\Omega sin(\phi)$ is the Corliolis parameter which represents the component of $\boldsymbol{\Omega}$ in $\boldsymbol{k}$ direction of the local rectangular coordinate system, and $\boldsymbol{u_h}$ is the vector of the horizontal wind. .*

**Comment:** p6 line 196-197: I guess altitude is retrieved from the geopotential. Maybe state it explicitly.

**Reply to comment:** That is correct. We have added the according information in the methods description.

*p.7 L203: The altitude at each model level is derived from the geopotential after vertically integrating the hydrostatic equation from the pressure and temperature profiles (for further information refer to the IFS documentation, ECMWF (2016)).*

**Comment:** P7 l 211: This paper is submitted within a special issue (WISE) and I guess the field campaign motivated the choice of the date. This could be mentioned here.

**Reply to comment:** We have added this information.

*p.8 L222: This study was performed in the context of the airborne research campaign WISE that took place during SON 2017 over the North Atlantic, which motivated the choice of the date.*

**Comment:** P10 line 251, figure 5a) and related discussion: Could you also show the equivalent of Fig. 4 a) on top of Fig 4c) which is shown here? This would emphasize the relevance of using tropopause relative coordinates and help understand lines 256-258.

**Reply to comment:** We have included the proposed subfigure along with the analysis relative to the CPT in Fig. 4 in the manuscript, respectively Fig. 2 in the present document.

*p.10 L262: Figure 5a shows the 10 year temporal and zonal average occurrence frequency for strong vertical wind shear*
215   $S \geq S_t^2$*, with the geometric altitude as the vertical coordinate. The mean LRT altitude for the same time period and region is indicated by the black solid line. Three distinct occurrence frequency maxima are apparent, i.e., in the midlatitudes between* $40° - 60°$ *N and mainly above the LRT, at the tropopause break at about* $30°$ *N above and below the LRT, and in the tropics in close vicinity above and below the LRT. The rearrangement of the grid boxes in the LRT-relative vertical coordinate system (Fig. 5b) concentrates the occurrence frequency maxima in a distinct layer above the LRT. This layer spans from the tropics to*
220   *latitudes north of* $60°$ *N, and it exhibits occurrence frequencies of the order of* $1\,\%$*–*$10\,\%$ *over a vertical range of about* $1 - 2$ *km.*

**Comment:** p 13 line 305-306 and Fig. 6: Do you know how exactly this surface is defined in the ECMWF ? In particular, I am surprised that the PV=2 PVU surface crosses the equator in Fig. 6. If there is some adjustment at low latitudes in the ECMWF field it would be useful to mention it here.

225   **Reply to comment:** The dynamic tropopause provided by the ECMWF is defined as the $Q = 2$ pvu isosurface or the 96 hPa if $Q = 2$ pvu is located at lower atmospheric pressure. We have now excluded the dynamic tropopause information for tropical latitudes throughout the manuscript to prevent misconceptions, since the "dynamic tropopause in the tropics" was never directly used in the analysis.

230   **Comment:** P 18 l 384: low $\longrightarrow$ lower . 40 m/s is not a particularly low wind speed even compared to the subtropical or eddy-driven jet.

**Reply to comment:** You are correct. Nevertheless, our intent was to emphasize that frequent strong wind shear is not necessarily linked to large jet core wind speeds. We have changed the phrasing and added a reference to the STJ core speeds identified by Wu and Sun (2017).

235

*p.20 L440: They exhibit a limited vertical extent and maximum wind speeds around 40 ms$^{-1}$, which is low e.g. compared to the EAJS core speeds identified by Wu and Sun (2017))..*

**Comment:** P 18 l 391: I am not sure how the geographic distribution here can be compared with the radiosondes from 2
240   stations in Sunilkumar et al. (2015). Agreed, the stations are influenced by the TEJ but from two points it seems complicated to validate a geographic pattern. A slight difference is that S2015 see this increase above the monthly mean CPT (their fig. 4), which might be 600 m (2-3 ERA5 levels) above the lapse rate (Sunilkumar et al., 2013; Munchak and Pan, 2014). Given the depth of the layer (1 km) used by the authors to investigate shear in tropopause relative coordinates, 600 m is significant. See also main comment 3.

245   **Reply to comment:** We agree that this sentence was misleading. We reversed the sentence to make the statement more clear. The intention was to put the results in context with observational studies in this region, since the occurrence frequency maximum over the Indian Ocean is very striking and the occurrence of strong shear is not directly intuitive. Concerning the second part of your comment, we agree that the TSL depth of 1 km was set to low to investigate the geographic mapping of

strong wind shear above the LRT, particularly in the tropics. We changed the search radius for the criterion of strong wind shear occurrence near to the tropopause to the region from 1 km below the LRT to 2 km above. This applies to all analysis steps throughout the manuscript. This change has no implications on the interpretation of any of the results, only the occurrence frequencies are generally shifted towards slightly larger values, particularly in the tropics. Thus, the analysis is more consistent.

*p.20 L446: Furthermore, the occurrence frequencies for strong vertical wind shear which were derived from radiosonde measurements by Sunilkumar et al. (2015) agree qualitatively with the ones at the respective geographic location of each radiosonde station in Fig. 7c.*

**Comment:** p 19 l 409 : Could you provide the correlation coefficient ?

**Reply to comment:** We have included the correlation coefficient.

*p.22 L466: The comparison of these time- and area-averaged frequencies with the Oceanic Niño sea surface temperature anomaly Index values for DJF (ONI) show an anticorrelation (Fig. 13a), with a Pearson correlation coefficient of $r = -0.788$.*

**Comment:** P 20 l 410: 'neutral and La Nina conditions': do you mean ' neutral and El Nino conditions'?

**Reply to comment:** The time and area averaged occurrence frequencies maximise along with negative ONI values which indicate la Niña phases.

**Comment:** P 20 l 440: you might consider showing a scatter plot of N2 and S2 to demonstrate this

**Reply to comment:** We agree that this statement concerning the non-correlation of $N^2$ and $S^2$ needs to be substantiated. The discussion section now includes such a 2d-histogram (Fig 3a in the present document, Fig. 14a in the manuscript), along with the according discussion. This figure is also used in the subsequent discussion on the occurrence of comparatively low Richardson numbers in the lower stratosphere.

*p.23 L499: Figure 14a shows the relative occurrence frequency of $N^2$-$S^2$ pairs for all ten years and in the region between the LRT and 3 km above. The majority of grid volumes exhibit a static stability between the stratospheric background $\overline{N^2}_{strat.} = 4 \cdot 10^{-4}\ \mathrm{s}^{-2}$ and moderately enhanced values associated with the TIL. At the same time, comparatively "weak" vertical wind shear $S^2 < 4 \cdot 10^{-4}\ \mathrm{s}^{-2}$ is most prevalent. Vertical wind shear and static stability do not correlate, and enhanced values of $S^2$ can be found within a large spectrum of $N^2$ under the condition that dynamic stability $Ri > Ri_c$ is (for the most part) maintained. Particularly the largest values of $N^2$ and $S^2$ do not correlate..*

**Comment:** p 21 l 450: you might note that ERA 5 has been shown to represent realistically part of the gravity wave activity (e.g., Krisch et al., 2020; Podglajen et al., 2020), which justifies that Gws might indeed be responsible for the enhanced shear in the reanalysis.

[Figure]

**Figure 3.** a) Relative occurrence frequency distribution of $N^2$-$S^2$ pairs in the region between the LRT and 3 km above, for all daily northern hemispheric ERA5 fields from 2008-2017. Logarithmic occurrence frequency color scale. Red dashed line indicates $S^2 = S_t^2$. Dashed black lines indicate the Richardson numbers 0.25, 0.5, 1.0 ,2.0 and 4.0. b) Histogram of the relative distribution of Richardson numbers associated with the data displayed in panel a. Orange bars show $Ri$ for grid volumes with $S^2 \geq S_t^2$, and blue bars for the remaining grid volumes between the LRT and 3 km above. Dotted black line indicates $Ri = 1$.

**Reply to comment:** Thank you for referencing these research studies, they helped to put the results better into context. We cited both studies in the discussion section, along with the recent research study on vertical wind shear in the IFS and the UTLS region by Schäfler et al. (2020).

*p.24 L539: Recently, Podglajen et al. (2020) compared long-duration superpressure balloon measurements with Lagrangian trajectories calculated from a set of numerical reanalysis products, and where able to show that the ERA5 reanalysis resolves central features of the gravity wave spectrum. The underlaying IFS model resolves low frequency large scale gravity waves down to wavelengths which approach the effective resolution, which is generally estimated to exhibit a factor of about 10 compared to the effective grid spacing. The assimilation of high resolution observational data further enhances the gravity wave activity in the model, which likely involves generation processes and gravity wave scales that are not resolved in the IFS. Furthermore, Krisch et al. (2020) identified individual wave packets in the ERA5 data that had been observed with the Gimballed Limb Observer for Radiance Imaging of the Atmosphere (GLORIA) during the GW-LCYCLE airborne measurement campaign. While these studies confirm that modern reanalysis products are capable to resolve central features of the gravity wave spectrum, the overall vertical wind variability due to gravity waves is likely still underestimated in the ERA5. Recently, Schäfler et al. (2020) reported a significant underestimation of the vertical wind shear near to the tropopause in the IFS, based on a comparison of Doppler wind lidar measurements with IFS analysis and forecast data. The analysis and forecast errors were most prominent at elevated tropopause altitudes above upper tropospheric ridges, i.e., regions that contribute significantly to*

*the occurrence of the TSL in the extratropics. To put these results into context, the model version used by Schäfler et al. (2020) exhibits a spectral truncation of TCo1280 on 137 vertical level, thus, the same vertical grid spacing compared to the ERA5 as well as a larger horizontal resolution.*

305     **Comment:** p 21 l 459 and fig. 13: for comparison, you could depict the distribution of Ri for all values of shear in the same region as well as the distribution over a deeper layer, to determine whether or not the TIL is a region of low Ri number

    **Reply to comment:** Such a histogram is now included in panel b of Fig. 14 in the manuscript (Fig. 3b in the present document). It supports the remarks that are made in the discussion section concerning the dynamic stability in the lower stratosphere in general and in the TSL.

310

    *p.25 L557: The grid volumes with $S^2 < 4 \cdot 10^{-4}$ s$^{-2}$ in Fig. 14a are mostly located below the diagonal dashed black line which indicates Richardson numbers of $Ri = 1$. The corresponding relative distribution of Richardson numbers is shown in Fig. 14b. The distribution peaks at $Ri = 3$ and spans over a large spectrum of larger Richardson numbers (only a section of the distribution is displayed). Richardson numbers of $Ri < 1$ are rarely associated with vertical wind shear $S^2 < 4 \cdot 10^{-4}$ s$^{-2}$.*

315 *This indicates that the lowermost stratosphere up to 3 km above the LRT is dynamically stable in the absence of strong vertical wind shear. In contrast, a significant proportion of grid volumes which exceed $S_t^2$ in Fig. 14a are located above the $Ri = 1$ isoline. These grid volumes constitute the greater part of Richardson numbers $Ri < 1$ within the first 3 km above the LRT, which is indicated in Fig. 14b.*

320     **Typos and suggested reformulations:**

    **Comment:** p 2 l 30: "an substantial" $\longrightarrow$ a substantial

**Reply to comment:** Done.

    **Comment:** p 2 l 37: 'linear wave theory' $\longrightarrow$ 'linear theory'

325 **Reply to comment:** Done.

    **Comment:** p 6 l 185 : I think it is the pressure velocity $\omega$ rather than $w$ which is provided by ECMWF.

**Reply to comment:** Yes, thank you for the hint. We changed the variable name accordingly.

    **Comment:** P10 l 255 : 'compare e.g.' $\longrightarrow$ 'compare with'

**Reply to comment:** Done.

330     **Comment:** p 13 l 298 : 'barclinic'

**Reply to comment:** Done.

    **Comment:** p 24 l 547: 'fulfils' $\longrightarrow$ 'fulfills'

**Reply to comment:** Done.

    **Comment:** p 18 legend of Fig. 11: "destails"

335 **Reply to comment:** Done.

**Comment:** p 22 l 467: "e.g." should be before the reference

**Reply to comment:** Thank you for pointing out the error.

**Comment:** p 24 l 544: I would remove 'exceptionally' since your analysis shows that this feature is not an exception

**Reply to comment:** We agree. We changed the phrasing thoughout the manuscript, also motivated by a similar comment by the other reviwer.

**Comment:** p 24 l 552 : operational analysis $\longrightarrow$ ERA 5

**Reply to comment:** Yes indeed, thank you!

**Point-by-point response to the second review**

**Main comments:**
* * *
**Comment 1:** Generally, the existence of wind shear above the jet core and in the lower stratosphere is not a surprise as is expected from balanced dynamics, the exact structure of the shear zones however are more involved. The authors mention the relation of the shear layer to the thermal wind balance at several instances in the manuscript along with other mechanisms. How much of the structure of the shear layer can be explained by the thermal wind relation? It should be possible to quantify this based on the ERA5 fields. The possible role of gravity waves is mentioned in several sections and maybe this way the magnitude of their contribution could be narrowed down. Furthermore, I would suggest to emphasize more clearly in the introduction, perhaps in a single summarizing sentence, what the main unknown aspects of the shear layer are (e.g. detailed structure, strength, vertical extent and occurrence in a statistical sense, formation mechanism) and which of these aspects are addressed in the study.

**Reply to comment 1:** We thank the reviewer for this suggestion. We started to compare the wind shear based on the full model wind and the thermal (model) wind. We extended the discussion in Sect. 3 to include a comparison of these two metrics based on a single day analysis (Fig.15 in the manuscript, Fig. 1 in the this reply). The comparison shows that there is in general a good agreement in the larger scale in the extratropics, however, with distinct differences which are presumably mainly caused by resolved gravity waves. We have not included a climatological analysis of this comparison. We think that such a comparison would increase the content of this manuscript substantially. Also new metrics would need to be introduced, since the metrics used in this study are not suited for such a comparison. The zonal averaging of occurrence frequencies of $S^2 \geq S_t^2$ as well as the quasi-horizontal mapping of strong wind shear near to the tropopause could show a good agreement between actual wind shear and thermal wind shear, even if the thermal wind does not represent the shear regions well, due to a superposition of underestimation and overestimation of the actual wind shear at different longitudes. This is indicated in the exemplary cross sections. We motivate further research on a more general comparison of the TSL with the thermal wind relation at the end of the paragraph.

Concerning the second part of your comment, we have included a more detailed description of the aim of this study at the end of the discussion section.

_p.24 L514: The connection between the temperature field and the vertical wind shear for synoptic scale flow can be approximated through the thermal wind relation, i.e., the vertical gradient of the geostrophic wind under the assumption of hydrostatic balance. Figure 14a shows the geostrophic wind at 200 hPa on 11 September 2017, i.e., the date of the exemplary single day analysis in Sect. 3. Overall, the geostrophic wind approximates the synoptic scale flow realistically (compare Fig. 2). However,_

*it overestimates the absolute zonal wind speed in cyclonic rotational systems like the one over the northwest Atlantic, in ac-*

380 *cordance with the fact that inertial forces are neglected (Holton and Hakim, 2012). The vertical structure of the geostrophic wind is shown exemplarily in the vertical cross sections at $60°$ W (Fig. 14b and c). The thermal wind relation results in several regions of strong vertical wind shear near to the tropopause. The comparsion with the vertical wind shear derived from the full model wind reveals a certain degree of agreement, in particular on the synoptic scale, but also differences on the smaller scales. The strength of the vertical wind shear at $40°$ N and 15 km altitude is overestimated by the thermal wind approximation,*

385 *as well as the shear region directly above that reaches north- and downward. The southward extent of the region of strong wind shear on the other hand is underestimated. The two pronounced wind shear regions below the tropopause and south of $40°$ N are not evident in the the thermal wind shear. The geostrophic zonal wind in the upper troposphere at about $50°$ N deviates from the full model zonal wind, which results in a significant underestimation the vertical wind shear below the tropopause. At the same time, the thermal wind relation overestimates the shear region below the tropopause at $45°$ N, which is caused*

390 *by strong meridional temperature gradients (not shown). The maximum of the thermal wind shear at $45°$ N directly above the LRT is not evident in the full model wind shear, but instead is apparent in a region that is located further to the north. Overall, the comparison indicates the significance of dynamic processes on smaller scales on which other forces than pressure gradient and Coriolis force need to be taken into account (Newton and Persson, 1962). This example already shows that many details, especially related to mesoscale dynamic features, need to be considered to fully address the differences in the vertical wind*

395 *shear based on the full model winds and on the thermal wind relation. A comprehensive analysis of these differences is beyond the scope of the current study but will be pursued in future work.*

*p.5 L160:This work presents an approach towards such an analysis, on the basis of ten years of northern hemispheric ECMWF ERA5 reanalysis data. The goal is to present a consistent area-wide analysis of the vertical and geographic occur-*

400 *rence frequency distribution for strong wind shear in a state of the art long term numerical representation of the atmosphere, i.e., the ERA5 reanalysis. Compared to observational research studies our approach has no spatial limitiations to assess the occurrence frequency. However, for our analysis it is necessary to keep in mind that the vertical wind shear features are only as well represented as the model resolution allows them to be. An important factor in this context is the vertical resolution, which has improved significantly compared to the ERA-Interim reanalysis as a reference dataset (e.g., Hoffmann et al., 2019).*

405 *Another important aspect is the choice of the central analysis method, i.e., the analysis of the occurrence frequency distribution of vertical wind shear above a certain threshold value. This approach conserves more information on where strong wind shear occurs in exchange for a loss of information concerning the actual strength of the wind shear, in contrast to the more common approach of tropopause based averaging.*

410 ________________________

**Comment 2:** I think the authors should reconsider some expressions and definitions related to the shear layer phenomenon.
- The words "enhanced" or "exceptional" are used frequently. In what sense is the wind shear "enhanced", compared to what

reference? The study shows that the layers of strong wind shear above the tropopause occur rather frequently and strong wind shear is certainly not exceptional near jet streaks.

415     - The tropopause shear layer (TSL) is defined based on an occurrence frequency criterion. In this sense, it is a purely statistical feature. Since a layer of strong wind shear also seems to be physically present and nicely visible in instantaneous synoptic situations with a strong jet stream (see Fig. 13b), I find it unfortunate to define the "TSL" in a statistical sense rather than as a synoptic feature. It would be more intuitive to call the regions indicated by the red contours in Fig. 13b "TSL".

420     **Reply to comment 2:** Concerning the first part of your comment, thank your for pointing that out, we agree that the wording was inconclusive. We changed the wording consistently to "strong vertical wind shear" throughout the manuscript, including the title of the study.

    Concerning the second part of your comment, we agree that the choice to define the TSL in such a way is controvertible. However, we wanted to distinguish the individual shear regions from the layer that emerges as an occurrence frequency maximum in the zonal and/or temporal mean. The idea behind the current definition of the TSL was to avoid a definition as to what exactly qualifies as a tropopause wind shear layer in instantaneous wind fields. This concerns for example the horizontal extent, which would raise the question if e.g. a shear region with horizontal extent of the order of 10 km should be defined as a shear layer or rather part of a shear layer. Similarly, this concerns the underlaying forcing mechanism, e.g., does a mesoscale gravity wave perturbation with a sequence of $S^2$ maxima and minima at the tropopause define individual shear layers or does it rather contribute to the overall occurrence frequency maximum of strong wind shear at the tropopause. We agree with your concern, however, we prefer to keep the current definition of the TSL. We have extended the motivation for the choice of the TSL definition in the manuscript.

435     *p.11 L270: In the following we will refer to the feature of maximum occurrence frequencies at the LRT in the zonal and/or temporal mean as a tropopause shear layer (TSL), however, a comparison with the TIL should be made cautiously. Both features appear similarly in tropopause-relative zonal means (compare with Zhang et al. (2019)), however, the wind shear layer emerges less frequently as well as less area-wide. Furthermore, a different metric is applied here, which analyses the tropopause-relative occurrence frequency of a threshold value $S_t^2$, instead of directly averaging $S^2$. We refrain from referring to individual regions of strong wind shear at individual time steps as a tropopause wind shear layer, because this would*
440     *raise the question of a lower limit for the horizontal and temporal scales that mark such a layer, considering the pronounced mesoscale variability of these regions. We realise that this choice is controvertible, because on a mesoscale horizontal extent of the order of 100 km the geometric aspect ratio between horizontal and vertical extent still clearly describes a layer-like character.*

445 ________________

    **Comment 3:** The authors have chosen a well-considered threshold $S_t^2$ and the choice is sufficiently explained. However, it would be interesting to test how sensitive the results are with respect to the threshold. How would the pattern of the occurrence

frequencies change if $S_t^2$ was even higher?

450     **Reply to comment 3:** We have repeated the analysis concerning the vertical distribution of grid volumes in the different vertical coordinates for a larger threshold value of $S_{t2}^2 = 6 \cdot 10^{-4} \, \mathrm{s}^{-2}$ (Fig. 2 in the present document). While these wind shear values occur significantly less frequently, the LRT-relative averaging still results in the occurrence of a TSL with 10 year temporal and zonal averaged occurrence frequencies of about $1 - 5$ %. Overall, the analysis matches the results of the one with $S_t^2 = 4 \cdot 10^{-4} \, \mathrm{s}^{-2}$, with the main difference being generally reduced occurrence frequencies. This applies also to the analysis

455     in the vertical coordinates relative to the CPT and the dynamic tropopause (associated with your comment 4 and comment 3 in the first review). These results are interesting, however, we prefer to not include this analysis in the manuscript, to keep the focus on the threshold that has been motivated in the introduction and the methods section.
* * *
460     **Comment 4:** I would be curious if the statistical analysis (for the midlatitudes) has also been done relative to the dynamical tropopause and whether there are any differences compared to the LRT-relative framework. This would be interesting e.g. in the context of many STE studies which focus on transport across the 2-PVU surface.

        **Reply to comment 4:** Motivated by his comment and a similar comment by the other referee concerning the cold point

465     tropopause, we repeated the analysis on the vertical occurrence frequency distribution in a vertical coordinate based on the distance of each grid volume from the $Q = 2$ pvu dynmic tropopause in the extratropics (Fig. 2c in the present document, respectively Fig. 5c in the manuscript). We discuss the similarities and differences compared to the LRT-realtive analysis in Sect. 4.1.

470     *p.13 L307: The significance of the processes which result in the occurrence of the TSL remains to be quantified. The clustering of grid volumes which exhibit $S^2 \geq S_t^2$ directly above the LRT in Fig. 5b agrees with the dynamic stability criterion and the thermal wind shear forcing associated with upper tropospheric fronts. However, the overall link between the tropopause definition, which is goverend by the temperature profile, and the occurrence of strong vertical wind shear remains uncertain. Therefore, we repeat the analysis for the tropical cold point tropopause (CPT), i.e., the absolute temperature minimum in the*

475     *tropical UTLS, and for the dynamic tropopopause, i.e., the $Q = 2$ pvu isosurface in the extratropics. The tropics feature a distinct separation of up to 1 km between the LRT and the CPT (Seidel et al., 2001), which motivates the comparison at low latitudes. The PV on the other hand does not constitute a useful tropopause definition in the tropics (Holton, 1995), which is why the dynamic tropopause is only used in the extratropics in this study. From Fig. 5c it is evident that ERA5 resolves the separation of the LRT and the CPT in the tropics, along with central features like a decreasing mean distance between the two*

480     *tropopauses towards the equator (Seidel et al., 2001). The clustering of strong wind shear grid volumes above the CPT is less pronounced compared to the LRT, along with a more pronounced secondary maximum below the CPT, which indicates that the occurrence of strong vertical wind shear is more closely linked to the LRT in the tropics. In the extratropics, the analysis in*

*a vertical coordiante system relative to the dynamic tropopause (Fig. 5c and north of $20°$ N) compares well with the analysis in the LRT-relative vertical coordinates. At high latitudes, the profiles which exhibit strong vertical wind shear are associated*

*with above average dynamic tropopause altitudes. However, the clustering of grid volumes with $S^2 \geq S_t^2$ above the dynamic tropopause exhibits a larger vertical spread, particularly above the tropopause break. The dynamic tropopause is identified systematically below the LRT in this region, due to the inclusion of PV streamers, as indicated in Fig. 5. Therefore, a larger amount of grid volumes which exhibit strong wind shear are shifted upwards during the averaging process. In conclusion, the averaging of the grid volumes which exhibit $S^2 \geq S_t^2$ based on the distance from the different reference altitudes reveals that*

*the elevation of strong vertical wind shear is (not generally but in the overall mean) more closely linked to the LRT than to the CPT or the dynamic tropopause.*
* * *
**Comment 5:** The introduction is quite long, the authors might consider shorten it a bit if possible.

**Reply to comment 5:** This is a valid point, and we tried to shorten the introduction. However, the information on the subject of strong wind shear in the tropopause region is scattered over a wide range of research studies, which results in the rather extensive introduction section. The following paragraphs have been removed because they did not add relevant information to the introduction. Particularly the third paragraph anticipated results from the analysis sections, and thus, is removed to avoid repitition:

*p.3 L69: The results compare well with the ones from Munich, exhibiting a more pronounced averaged vertical wind shear peak during winter, as well as a larger vertical spread compared to summer.*

*p.3 L84: even though the authors did not make use of a tropopause-relative vertical coordinate.*

*p.4 L118: This fact, however, is not reflected in the occurrence frequency of reduced Richardson numbers or enhanced turbulence index ($TI$) values at tropopause altitudes (Jaeger and Sprenger, 2007), which might indicate that the wind shear near to the tropopause is not as pronounced compared to other jet streams. It should, however, be considered that neither the Richardson number nor the $TI$ are solely defined by $S^2$. The occurrence of a vertical wind shear peak near to the tropopause, as it is apparent in the observational studies, is in fact not necessarily linked to exceptionally large wind speed, because of the limited vertical extent of the shear regions as well as due to the fact that directional shear can contribute to the total wind shear. The summer TEJ presents a descriptive example for this issue. The upper-tropospheric easterlies which define the TEJ exhibit average wind speeds around 40 m/s, which is rather slow compared to the winter STJ and polar jet. They are however associated with the most pronounced near-tropopause maximum in $S^2$ (Sunilkumar et al., 2015; Zhang et al., 2019)*
* * *
**Specific comments, suggestions and typos:**

520    **Comment:** L13ff: Throughout the manuscript, the term "tropopause-based" vertical wind shear is used frequently. This expression is not very clear to me; does it mean "tropopauserelative", "near-tropopause" or "tropopause-level"?).

    **Reply to comment:** We agree and changed the wording consistently to "strong vertical wind shear near to the tropopause".

    **Comment:** L30: an –> a

525    **Reply to comment:** Done.

    **Comment:** L36: to the –> its

    **Reply to comment:** Done.

530    **Comment:** L60 and all following occurrences: ° N –> °N, please remove the space between "°" and "N"/"E"/"S"/"W"

    **Reply to comment:** The WCD guidelines state that "coordinates need a degree sign and a space when naming the direction (e.g. 30° N, 25° E)."

    **Comment:** L61: for –> of

535    **Reply to comment:** Done.

    **Comment:** L67: ERA Interim –> ERA-Interim

    **Reply to comment:** Done, and in one other instance. Thank you for pointing that out.

540    **Comment:** L67: data set –> dataset

    **Reply to comment:** Done.

    **Comment:** L70: remove "on"

    **Reply to comment:** Done.

545

    **Comment:** L93: data –> forecasts

    **Reply to comment:** Done.

    **Comment:** L98: analysis data –> analyses

550    **Reply to comment:** Kaluza et al. (2019) use operational analysis data from the ECMWF IFS.

    **Comment:** L105: presents –> constitutes

**Reply to comment:** Done, good suggestion.

555     **Comment:** L109: causes –> cause

    **Reply to comment:** Done.

    **Comment:** L173: analysis "of" a single day

    **Reply to comment:** Done.

560

    **Comment:** L174: Sections –> Section

    **Reply to comment:** Done.

    **Comment:** L178: Please check the date and time convections of WCD

565     **Reply to comment:** The WCD guidelines state: "Date and time: 25 July 2007 (dd month yyyy)". We now use this convention consitently throughout the manuscript.

    **Comment:** L185: ERA5 provides omega in Pa/s, not w in m/s

    **Reply to comment:** Thank your for pointing that out. We changed the variable name accordingly.

570

    **Comment:** L189: dynamic –> dynamical

    **Reply to comment:** Done.

    **Comment:** L191: Please remove the symbol for the cross product and insert a comma after equation. / L192: (with the

575 angular velocity of the Earth, Omega).

    **Reply to comment:** Done.

*p.6 L191: Following the definition of Ertel (1942) the potential vorticity (PV) can be written as*

$$Q = \frac{1}{\rho}\boldsymbol{\eta} \cdot \boldsymbol{\nabla}\Theta, \tag{3}$$

580 *where $\rho$ is the density of the medium, and $\boldsymbol{\eta} = \boldsymbol{\nabla} \times \boldsymbol{u} + 2\boldsymbol{\Omega}$ the vector of the absolute vorticity with the angular velocity of the*

*earth $\boldsymbol{\Omega}$.*

    **Comment:** L196: How do you derive the vertical distance between the model levels, do you use geopotential?

    **Reply to comment:** That is correct. We have added the according information in the methods description

585

[Figure]

**Figure 4.** Vertical cross sections on 11 September 2017, at 00 UTC and at a) 120° W, b) 60° W, and c) 100° E. Colour contour shows zonal wind speed ($u$, in m s$^{-1}$), black dots LRT altitude, and black dotted lines isentropes ($\Theta$, in K). Magenta lines show $S^2 = S_t^2$ isolines.

*p.7 L203: The altitude at each model level is derived from the geopotential after vertically integrating the hydrostatic equation from the pressure and temperature profiles (for further information refer to the IFS documentation, ECMWF (2016)).*

**Comment:** L201: can not –> cannot

**Reply to comment:** Done.

**Comment:** L202: for –> to

**Reply to comment:** Done.

**Comment:** L203: It is a bit confusing to read about static stability in combination with the notation $S_t^2$.

**Reply to comment:** Thank your for pointing that out, it was a typo.

**Comment:** L209: majorly –> mostly

**Reply to comment:** Done.

**Comment:** Fig3: It would be interesting to see contours of $S^2$ in the snapshot vertical cross sections in addition to wind speed. This would illustrate not only the general position of the shear zones but also the spatial variability.

**Reply to comment:** We have included isolines that indicate the location of $S^2 = S_t^2$ in the vertical cross sections (Fig. 4 in the present document, Fig. 3 in the manuscript). We furthermore replaced the cross section in Fig. 3b with the cross section at 60° W, which was previously shown in Fig. 13b. Thus, the cross section is not implemented twice.

**Comment:** L259-264: Here, it would be helpful to explicitly point out the different positions of the solid/dashed black lines in Fig. 5a.

**Reply to comment:** We agree, and we have now pointed out the different tropopause averages and the according lines in Fig.

610    5 in the paragraph that discusses the vertical cross sections. Furthermore, the depiction of the different tropopause averages is now more consistent throughout the manuscript (Fig. 4, 5 and 11)

**Comment:** L267-270: While the schematic illustration in Fig. 5b is very helpful and easy to understand, I find the explanation in the text rather unclear. The authors might consider rewriting these sentences.

615    **Reply to comment:** We extended the description of the averaging effect that results in the large spread of the secondary occurrence frequency maxima above and below the tropopause break.

*p.12 L288: Furthermore, an enhanced vertical spread of $S^2 \geq S_t^2$ is apparent at the tropopause break in the LRT-relative vertical coordinates (Fig. 5b) compared to the geometric altitude (Fig. 5a). Figure 6 illustrates how this can be explained by*

620    *the averaging method. Below the jet core of the STJ strong vertical wind shear occurs frequently within PV streamers (Škerlak et al., 2015), i.e., tounges of stratospheric air that reach south- and downward. They are characterised by stratospheric PV values, which is linked to the frontal character of the inherent temperature gradients. The horizontal temperature gradients are associated with a strong thermal wind forcing, and the resulting vertical wind shear can be sustained by the vertical temperature gradients. Despite the stratospheric characteristics of these air masses, the LRT criterion which requires a mean lapse rate*

625    *below 2.0 K km$^{-1}$ over a vertical distance of 2 km is no longer met at some point within the PV streamers, which is indicated by the dashed black line in Fig. 6. In these cases the LRT is located at the upper edge the tropopause break and several kilometers above the region of strong wind shear. Eventually, the 10 year temporally and zonally averaged LRT altitude exhibits a more smooth transition between the lower and the upper edge of the tropopause break (right panel in Fig. 6), which is caused by short-period as well as seasonal meandering of the tropopause break. The LRT-relative averaging method ultimately shifts the*

630    *region of strong wind shear downward from the mean LRT altitude, over the distance which was originally determined relative to the instantaneous LRT. The equivalent effect also occurs at the upper edge of the tropopause break as indicated in Fig. 6.*

**Comment:** L298: barclinic –> baroclinic

**Reply to comment:** Done.

635

**Comment:** L315-316: I assume your background state is still latitude-dependent? From this sentence it is not clear if you also average over latitudes.

**Reply to comment:** This information is now explicitly stated.

640    *p.15 L371: We preserve the meridional dependency and define the zonal and temporal average $\overline{\Theta}(Q = 2 \,\mathrm{pvu})$ in the meridional region from $35°$ N–$60°$ N as the background state for the following analysis.*

**Comment:** Fig7a: What does the black dot indicate?

**Reply to comment:** The black dot indicates $\overline{\Theta}(Q = 2\text{ pvu})$ at $51°$ N and during SON over the Atlantic region, i.e., the mean potential temperature that defines the background state for the analysis in Fig. 9. This information was missing in the caption of Fig. 7 and is now included.

**Comment:** L326: Why did you choose exactly 51°N?

**Reply to comment:** To some extent the choice is arbitrary, the primary intention was to introduce the metric. According to Fig. 10a the occurrence frequencies maximise at these latitudes. Thus, the choice represent a descriptive example.

**Comment:** L355: DFJ –> DJF

**Reply to comment:** Done.

**Comment:** L416: and "references" therein

**Reply to comment:** Done.

**Comment:** L440-441: No co-location of TIL and TSL: Can you show this in a figure? Maybe a snapshot vertical cross section would do. Or perhaps something like a frequency distribution of $N^2$ in the lowest 2 km above the LRT, showing grid cells with $S^2 \geq S_t^2$ separately and comparing them to the $N^2$ distribution of all grid cells.

**Reply to comment:** We agree that this statement concerning the non-correlation of $N^2$ and $S^2$ needs to be substantiated. The discussion section now includes such a 2d-histogram (Fig 3a in the present document, Fig. 14a in the manuscript), along with the according discussion. This figure is also used in the subsequent discussion on the occurrence of comparatively low Richardson numbers in the lower stratosphere.

*p.23 L499: Figure 14a shows the relative occurrence frequency of $N^2$-$S^2$ pairs for all ten years and in the region between the LRT and 3 km above. The majority of grid volumes exhibit a static stability between the stratospheric background $\overline{N^2}_{strat.} = 4 \cdot 10^{-4}\text{ s}^{-2}$ and moderately enhanced values associated with the TIL. At the same time, comparatively "weak" vertical wind shear $S^2 < 4 \cdot 10^{-4}\text{ s}^{-2}$ is most prevalent. Vertical wind shear and static stability do not correlate, and enhanced values of $S^2$ can be found within a large spectrum of $N^2$ under the condition that dynamic stability $Ri > Ri_c$ is (for the most part) maintained. Particularly the largest values of $N^2$ and $S^2$ do not correlate.*

**Comment:** L449-453: Are gravity waves (partially) resolved in ERA5?

**Reply to comment:** The other referee pointed out two recent research studies on that matter, which are now cited in the discussion section. Central features of the gravity wave spectrum are in fact resolved in the ERA5 reanalysis.

*p.24 L539: Recently, Podglajen et al. (2020) compared long-duration superpressure balloon measurements with Lagrangian trajectories calculated from a set of numerical reanalysis products, and where able to show that the ERA5 reanalysis resolves*

*central features of the gravity wave spectrum. The underlaying IFS model resolves low frequency large scale gravity waves*
*down to wavelengths which approach the effective resolution, which is generally estimated to exhibit a factor of about 10 com-*
*pared to the effective grid spacing. The assimilation of high resolution observational data further enhances the gravity wave*
*activity in the model, which likely involves generation processes and gravity wave scales that are not resolved in the IFS. Fur-*
*thermore, Krisch et al. (2020) identified individual wave packets in the ERA5 data that had been observed with the Gimballed*
*Limb Observer for Radiance Imaging of the Atmosphere (GLORIA) during the GW-LCYCLE airborne measurement campaign.*
*While these studies confirm that modern reanalysis products are capable to resolve central features of the gravity wave spec-*
*trum, the overall vertical wind variability due to gravity waves is likely still underestimated in the ERA5. Recently, Schäfler*
*et al. (2020) reported a significant underestimation of the vertical wind shear near to the tropopause in the IFS, based on a*
*comparison of Doppler wind lidar measurements with IFS analysis and forecast data. The analysis and forecast errors were*
*most prominent at elevated tropopause altitudes above upper tropospheric ridges, i.e., regions that contribute significantly to*
*the occurrence of the TSL in the extratropics. To put these results into context, the model version used by Schäfler et al. (2020)*
*exhibits a spectral truncation of TCo1280 on 137 vertical level, thus, the same vertical grid spacing compared to the ERA5 as*
*well as a larger horizontal resolution.*

**Comment:** Fig13b: Do the black circles indicate the LRT?

**Reply to comment:** That is correct. This cross section and the according description is now implemented in Fig. 3b in the manuscript (Fig. 4b in the present document).

**Comment:** L487: I believe that throughout the summary the expressions "exceptionally pronounced/strong/enhanced vertical wind shear" and "enhanced tropopause-based vertical wind shear" are used synonymously. Perhaps consider using the same formulation throughout the paper (including introduction) to not confuse the reader.

**Reply to comment:** Thank you for pointing that out, we agree that the inconsistency is confusing to the reader. We changed the wording, and now refer to $S^2 \geq S_t^2$ as "strong vertical wind shear", which includes the title of the study.

**Comment:** L528: dynamic –> dynamical

**Reply to comment:** Done.

[Figure]

**Figure 5.** Comparison of Fig. 4 in the originally uploaded manuscript (top row) and the equivalent in the revised manuscript (bottom row). The colorbars had to be extended slightly at the top end due to the larger maximum values, however, with the same colors in the frequency range of $10^{-3} - 10^{-1}$.

**References**

[revised manuscript text omitted]

---

## Editor Decision (ED1)

wcd-2021-8
Editor decision – comments to the authors

**On the occurrence of strong vertical wind shear in the tropopause region: A ten year ERA5 northern hemispheric study**

by T. Kaluza et al.

Dear Thorsten

Many thanks for your revisions and for addressing the points raised by the reviewers in great detail. I am happy to accept your paper for publication in WCD subject to technical corrections, as listed below. In addition to these suggestions, I would like to invite you to consider shortening and/or simplifying the text in a few places. The text is very detailed and a bit heavy in some places. Some remarks appear to me as side remarks that maybe also could be omitted, for the benefit of an even better flow of the entire paper. I leave this to your discretion.

L6: $S_t^2$ is not yet defined, say that it refers to the squared vertical shear of the horizontal wind speed.

L6 and in several other places: "spectrum"? maybe better "distribution"? You later use "spectrum" for the wave or frequency spectrum (which I find fully OK), but then it is slightly confusing to me that the simple distribution of shear values is also called spectrum.

L15 and in other places: "near to the tropopause" → "near the tropopause"

L25: "la Nina" should read "La Nina"

L39: "… and the squared …"

L68: "expand … and present" should read "expanded … and presented"

L92: "describe" should read "described"

L107: not clear to me what "vertical and lateral shear zones" are. In both cases do you still mean vertical shear of the horizontal flow? So maybe you mean "… with shear zones vertically spanning … and laterally over several hundreds of kilometres"?

L120: "confirm" should read "confirmed"

L131: should read "orographically induced"

L145: should read "of small-scale perturbations"

L149: "analyse" should read "analysed"

L165: I suggest deleting "as a reference dataset"

L166: why "central"?

L167: The sentence "This approach …" is rather complicated. Can you say the same with simpler words?

L188: no need to repeat the WMO criteria, which you already listed in the introduction

Caption Fig. 1: why "tropospheric volume", why not just "Blue indicates values in the troposphere"?

L197: latex use backslash-sin for the sine symbol

L206: not sure that "larger resolved spectrum" is the right term here, maybe "in a bias towards larger values of vertical wind shear"?

L220: I don't understand "the mostly exclusive occurrence", maybe "the rare occurrence"?

L234: I think the "northwest Atlantic" should read "western North Atlantic"

L249: instead of "grid box volume" maybe simpler "… at least at one level between …"?

L264: delete "i.e.", or should it be "e.g."?

Caption Fig. 5: I don't understand "regions where negative (positive) vertical wind shear makes for 75 % of the counts"

L270: "at the LRT"? maybe better "near the LRT"?

L278: "controvertible"? I don't understand, do you maybe mean "controversial"?

L291/296: we don't call this a "PV streamer", rather a "tropopause fold". PV streamers are identified on isentropic PV charts, not in vertical cross-sections.

L291-294: I would delete these two sentences; I find them a bit shaky (what is "thermal wind forcing"?) and they are not really needed.

L296: "at the upper edge of the …"

L298: "more smooth" should read "smoother"

L324: I am not sure that the tropopause folds are the main reason for the fact that the 2-pvu tropopause is on average lower than the LRT (folds are relatively rare).

L340: "… winter, when it is located over the maritime …"

L345: "narrow down"? maybe better "discuss"?

L374 and in many other places: please check, but I think "the Northeast Pacific" should read "the eastern North Pacific" and likewise for Northwest Atlantic etc.

L378: "On the one hand …"

Caption Fig. 10: "location of Fig. 8c"? Maybe rather Fig. 9?

L450: not clear to me what "dz = 20 meter" means

L454: no need to introduce abbreviations (COT) that are not used later

L458: "over the maritime continent"

L463: you already introduced the abbreviation ENSO earlier

L472: why "responsible"?

Caption Fig. 14: panel a): the Ri contours should be labeled

L506: "in the North Atlantic …, Kaluza … described …"

L509: what do you mean by "an exclusive direct effect on either"? maybe "a contrasting effect on the two parameters"?

L526: typo "the the"

L527: "underestimation of the …"

L514-535: I don't understand the need for this paragraph. It reads like a maybe interesting side aspect, but it somehow distorts the flow of this discussion sector. I know you included this in a response to a reviewer comment, but for me it would be sufficient (or better) to include this paragraph only in the reply document.

L537: what are the dashed variables? And **k** should be bold. Not sure that you need here a mathematical notation because *T'* etc. are not mentioned again.

L544: I don't understand, how can gravity wave activity be enhanced in a model at a scale that is not resolved?

L554: do you mean "higher horizontal resolution"?

L629: El Nino and La Nina with capital E and L

L664ff (References): WCD uses journal abbreviations, e.g., "J. Geophys. Res." etc. Please check in published WCD papers and adjust the journal names.

I am looking forward to receiving the final version of your manuscript.
With best regards,
Heini

---

## Author Response (AR2)

**Authors response**

Dear Heini Wernli,

thank you for once again reviewing the document and for pointing out the remaining technical corrections. Furthermore, I understand and agree with your concern regarding the length and the reading flow of the paper, and I reworked or removed

5 a few sentences/paragraphs that did not add significant information to the paper. I hope that these changes along with your suggestion to not include the paragraph on the thermal wind shear improved the reading flow of the paper.
In the following I give a point-by-point response to each of your comments, in the order: 1) technical correction/comment, 2) our response and 3) the changes in the manuscript. The modified text passages are given in italics.

10

**Point-by-point response to the Co-Editors comments**

**Comment:** L6:  $S_t^2$  is not yet defined, say that it refers to the squared vertical shear of the horizontal wind speed.

15 **Reply to comment:** I added a description as to what  $S_t^2$  refers to.

**Changes in the manuscript:** L6: A threshold value of  $S_t^2 = 4 \cdot 10^{-4} \text{ s}^{-2}$  of the squared vertical shear of the horizontal wind is applied, which marks the top end of the distribution of atmospheric wind shear to focus on situations which cannot be sustained by the mean static stability in the troposphere according to linear theory.

20 **Comment:** L6 and in several other places: "spectrum"? maybe better "distribution"? You later use "spectrum" for the wave or frequency spectrum (which I find fully OK), but then it is slightly confusing to me that the simple distribution of shear values is also called spectrum.

**Reply to comment:** I agree that "distribution" is a better choice, and replaced the word "spectrum" in this context in several instances in the document.

25

**Comment:** L15 and in other places: "near to the tropopause"  $\rightarrow$  "near the tropopause" **Reply to comment:** Thank you for pointing that out, I changed the wording accordingly.

Comment: L25: "la Nina" should read "La Nina"

30 Reply to comment: Done.

**Comment:** L39: "... and the squared ..." **Reply to comment:** Done.

35 **Comment:** L68: "expand ... and present" should read "expanded ... and presented" **Reply to comment:** Done.

**Comment:** L92: "describe" should read "described" **Reply to comment:** Done.

40

**Comment:** L107: not clear to me what "vertical and lateral shear zones" are. In both cases do you still mean vertical shear of the horizontal flow? So maybe you mean "… with shear zones vertically spanning … and laterally over several hundreds of kilometres"?

Reply to comment: The intention was to describe wind shear surrounding the jet core in general, i.e., vertical shear of the

45 horizontal wind above and below the jet core over vertical distances of several kilometers, as well as the horizontal shear of the horizontal wind spanning laterally around the jet core over distances of several hundred kilometers. I tried to give a definition of the jet stream (dimension) based on the wind shear. But since the paper focusses on the vertical shear of the horizontal wind, I agree with your suggestion and changed the wording accordingly.

**Changes in the manuscript:** L106: *The jet streams constitute the planetary-scale background state for the distribution of wind shear, with shear zones vertically spanning over several kilometres and laterally over several hundreds of kilometres.*

**Comment:** L120: "confirm" should read "confirmed" **Reply to comment:** Done.

55 **Comment:** L131: should read "orographically induced" **Reply to comment:** Done.

**Comment:** L145: should read "of small-scale perturbations" **Reply to comment:** Done.

60

**Comment:** L149: "analyse" should read "analysed" **Reply to comment:** Done.

Comment: L165: I suggest deleting "as a reference dataset"

65 Reply to comment: Done.

Comment: L166: why "central"?

**Reply to comment:** I used this wording because the choice between the two common analysis methods, to either calculate the average of the vertical wind shear  $S^2$  or to calculate occurrence frequency distributions for a threshold determines key

aspects of what knowledge can be gained based on the analysis. The idea is to emphasize that the choice to analyse occurrence frequencies  $S^2 \ge S_t^2$  is central for the identification of the TSL. The changes in the manuscript motivated by your following comment should make that more clear.

**Comment:** L167: The sentence "This approach ..." is rather complicated. Can you say the same with simpler words? **Reply to comment:** Motivated by this comment and the previous one, I rephrased this description.

**Changes in the manuscript:** L165: Another important aspect is the choice of the central analysis method, where we compare two common approaches. The first approach is the tropopause-relative averaging of either the vertical wind shear  $S^2$  (e.g. Birner et al., 2002; Zhang et al., 2015) or of the horizontal wind and subsequent calculation of the mean vertical wind shear (Birner, 2006). The second approach and the one which is used in this work analyses the occurrence frequency distribution of

80  $S^2$  above a certain threshold value (Dvoskin and Sissenwine, 1958; Sunilkumar et al., 2015). This method can be advantageous because it conserves more information on where strong wind shear does (not) occur.

**Comment:** L188: no need to repeat the WMO criteria, which you already listed in the introduction **Reply to comment:** Thank you for pointing that out, I removed the description.

**85**

75

**Comment:** Caption Fig. 1: why "tropospheric volume", why not just "Blue indicates values in the troposphere"? **Reply to comment:** I agree with your suggestion and changed the caption accordingly.

**Changes in the manuscript:** Fig.1: *Blue indicates values in the troposphere* (z